# SparseDiT: Token Sparsification for Efficient Diffusion Transformer

**Shuning Chang**[1 2 3]   **Pichao Wang**[2*]   **Jiasheng Tang**[2 3]   **Fan Wang**[2 3]   **Yi Yang**[1]

[1]Zhejiang University    [2]Damo Academy, Alibaba Group    [3]Hupan Lab

shuning.csn@alibaba-inc.com

## Abstract

Diffusion Transformers (DiT) are renowned for their impressive generative performance; however, they are significantly constrained by considerable computational costs due to the quadratic complexity in self-attention and the extensive sampling steps required. While advancements have been made in expediting the sampling process, the underlying architectural inefficiencies within DiT remain underexplored. We introduce SparseDiT, a novel framework that implements token sparsification across spatial and temporal dimensions to enhance computational efficiency while preserving generative quality. Spatially, SparseDiT employs a tri-segment architecture that allocates token density based on feature requirements at each layer: Poolingformer in the bottom layers for efficient global feature extraction, Sparse-Dense Token Modules (SDTM) in the middle layers to balance global context with local detail, and dense tokens in the top layers to refine high-frequency details. Temporally, SparseDiT dynamically modulates token density across denoising stages, progressively increasing token count as finer details emerge in later timesteps. This synergy between SparseDiT's spatially adaptive architecture and its temporal pruning strategy enables a unified framework that balances efficiency and fidelity throughout the generation process. Our experiments demonstrate SparseDiT's effectiveness, achieving a 55% reduction in FLOPs and a 175% improvement in inference speed on DiT-XL with similar FID score on $512 \times 512$ ImageNet, a 56% reduction in FLOPs across video generation datasets, and a 69% improvement in inference speed on PixArt-$\alpha$ on text-to-image generation task with a 0.24 FID score decrease. SparseDiT provides a scalable solution for high-quality diffusion-based generation compatible with sampling optimization techniques. Code is available at https://github.com/changsn/SparseDiT.

## 1  Introduction

Diffusion models [18, 12, 43, 3] have emerged as powerful tools in visual generation, producing photorealistic images and videos by gradually refining structured content from noise. Leveraging the scalability of Transformers, Diffusion Transformer (DiT) models [41, 2] extend these capabilities to more complex and detailed tasks [13, 62] and form the backbone of advanced generation frameworks such as Sora [6]. Yet, despite their impressive generative capabilities, DiT models face significant computational limitations that restrict their broader applicability, especially in scenarios where efficiency is paramount.

The computational burden of DiT models arises mainly from the need for numerous sampling steps in the denoising process and the quadratic complexity of Transformer structures with respect to the number of tokens. Although recent research has focused on reducing this burden by accelerating the sampling process through techniques such as ODE solvers [51, 30], flow-based methods [26, 28],

---

*Work done at Alibaba Group, and now affiliated with Amazon.

39th Conference on Neural Information Processing Systems (NeurIPS 2025).

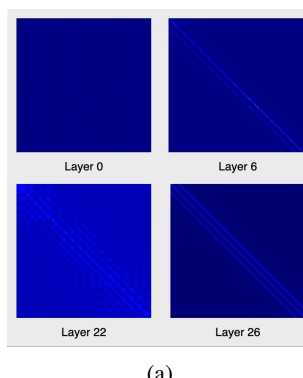

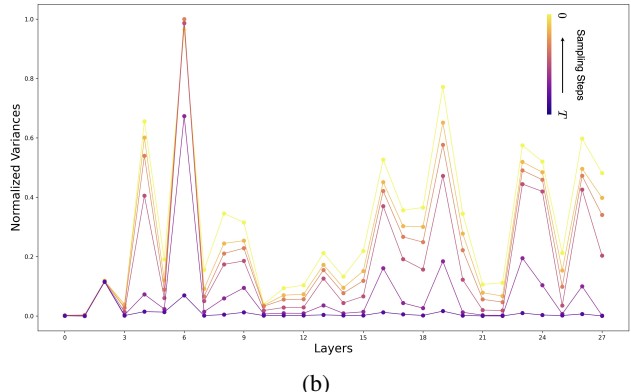

(a)                                    (b)

Figure 1: (a) The attention maps in different self-attention layers. Zoom-in for better visibility. (b) The normalized variances of attention maps in DiT-XL. Different curve lines represent different sampling steps.

and knowledge distillation [53, 47, 32, 35, 63], these approaches often overlook the core architectural inefficiencies of DiT itself. Unlike U-Net-based architectures [44], which mitigate complexity through a contracting-expansive structure, DiT's reliance on token-level self-attention incurs high costs that scale with model size and token count. This bottleneck highlights a critical need for DiT-specific innovations that manage token density intelligently, balancing efficiency and quality without sacrificing generation fidelity.

To address this challenge, we examine the distribution of attention across DiT's layers and sampling steps, seeking to uncover structural patterns that could guide more efficient token management. As illustrated in Figure 1a, attention maps reveal that tokens capture different levels of granularity across layers: in the bottom layers, such as layer 0, attention maps approximately appear uniform distribution, indicating a focus on broad, global features akin to global pooling. Meanwhile, middle layers alternate in their attention, with certain layers (*e.g.*, layer 6 and layer 26) capturing local details, while others (*e.g.*, layer 22) emphasize global structure. This observation is quantified in Figure 1b, where the variance in attention scores highlights DiT's consistent pattern of alternation between global and local feature extraction across layers. In addition to spatial analysis, Figure 1b also reveals insights across sampling steps. We observe that, while the pattern of alternation between global and local focus remains stable, attention variance increases as the denoising process advances, indicating a growing emphasis on local information at lower-noise stages. This analysis yields three core insights: (1) the initial self-attention layers exhibit minimal variance, showing low discrepancy in feature extraction across tokens; (2) DiT's architecture inherently alternates between global and local feature focus across layers, a pattern consistent across all sampling steps; and (3) as denoising progresses, the model increasingly prioritizes local details, dynamically adapting to heightened demands for detail at later stages.

Building on these insights, we propose SparseDiT, which reimagines token density management as a dynamic process adapting across both spatial layers and temporal stages. Spatially, shallow layers capture smooth, global features, making complex self-attention less efficient. Subsequent layers alternate between high-frequency local details and low-frequency global information, where sparse tokens efficiently capture global features, and dense tokens refine local details. Temporally, as denoising advances, the need for local detail increases, motivating a dynamic token adjustment strategy. This dual-layered adaptation integrates layer-wise token modulation—balancing efficiency and detail within each sampling step—and a timestep-wise pruning strategy that adjusts token density dynamically. Together, these spatial and temporal adaptations allow SparseDiT to achieve computational efficiency without sacrificing high-fidelity detail, as reflected in SparseDiT's architectural design and pruning strategy, tailored to meet the unique spatial and temporal demands of the generation process.

**SparseDiT Architecture**: SparseDiT's architecture employs a three-segment design that aligns token density with the features each layer captures. In the bottom layers, we replace self-attention with a poolingformer structure to capture broad global features through average pooling, reducing computation while preserving essential structure. In the middle layers, SparseDiT introduces Sparse-Dense Token Modules (SDTM), blending sparse tokens for global structure with dense tokens to refine local details. This design enables SparseDiT to retain efficiency while maintaining stability and

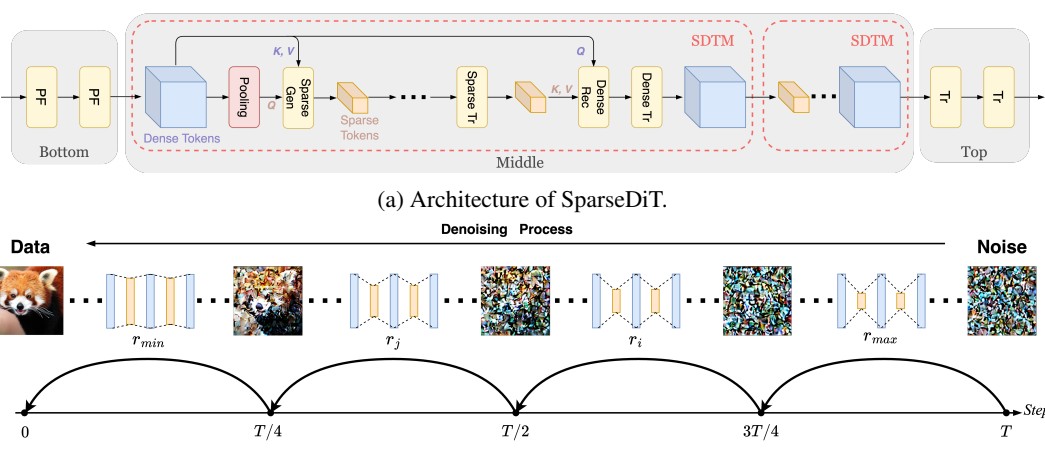

(a) Architecture of SparseDiT.

(b) Timestep-wise pruning rate strategy.

Figure 2: Architecture of SparseDiT and timestep-wise pruning rate strategy. The architecture consists of three segments: bottom, middle, and top. The bottom segment includes poolingformers (PF). The middle segment comprises multiple sparse-dense token modules (SDTMs), where sparse token generation transformers ("Sparse Gen") and dense token recovery transformers ("Dense Rec") alternate to balance global and detailed information processing. The top segment contains standard transformers ("Tr") for final processing. During the denoising process, we apply varying pruning rates $r$ at different stages (depicted as four stages).

detail. In the top layers, the model processes all tokens densely, focusing on high-frequency details for refined output quality.

**Timestep-Wise Pruning Rate Strategy**: Complementing spatial adjustments, SparseDiT's timestep-wise pruning rate strategy adapts token density across denoising stages. In early stages, where broad, low-frequency structures dominate, SparseDiT applies a high pruning rate, conserving resources. As the process progresses and intricate details emerge, the pruning rate decreases, allowing for a gradual increase in token density. This adjustment aligns computational effort with each stage's complexity, dynamically balancing efficiency and detail to optimize resources for high-fidelity output.

Our empirical results substantiate the effectiveness of SparseDiT's integrated approach. SparseDiT reduces the FLOPs of DiT-XL by 55% and improves inference speed by 175% on 512×512 ImageNet [11] images, with only a increase of 0.09 in FID score [17]. On the Latte-XL dataset [34], SparseDiT achieves a 56% reduction in FLOPs across standard video generation datasets, including FaceForensics [45], SkyTimelapse [60], UCF101 [54], and Taichi-HD [50]. Additionally, on the more challenging text-to-image generation task, we achieve a 69% improvement in inference speed on PixArt-$\alpha$ with a 0.24 FID score reduction. These results demonstrate SparseDiT's capability to provide an efficient, high-quality architecture for diffusion-based generation, compatible with further sampling optimization techniques for enhanced efficiency.

## 2 Related works

**Efficient sampling process.** This direction focuses on optimizing sampling steps in diffusion models. Some approaches [51, 30, 1, 31] design new solvers to achieve faster sampling with fewer steps. Consistency models (CMs) [53, 52, 32, 58, 21] are closely related to diffusion models, achieving by distilling pre-trained diffusion models or training from scratch. These models learn a one-step mapping between noise and data, and all the points of the sampling trajectory map to the same initial data point. Flow-based [26, 28, 72, 13, 71] approaches straighten the transport trajectories among different distributions and thereby realize efficient inference. Knowledge distillation are applied by some methods [53, 47, 32, 35, 63] to reduce the sampling steps and match the output of the combined conditional and unconditional models. Parallel sampling techniques [49, 70, 56] employ Picard-Lindelöf iteration or Fourier neural operators for solving SDE/ODE.

**Efficient model structure.** Another avenue focuses on improving inference time within a single model evaluation. Some methods attempt to compress the size of diffusion models via pruning [14]

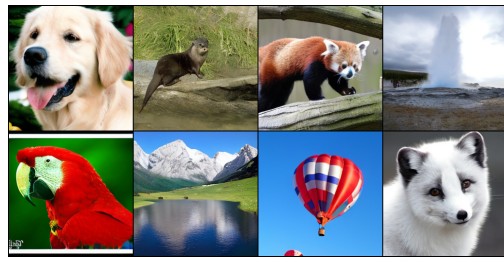
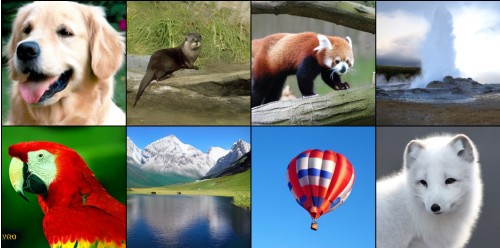

| (a) Standard DiT-XL. | (b) Modifying first two attention in DiT-XL. |

Figure 3: Comparison of images generated by the standard DiT-XL model and a modified DiT-XL model where the first two attention maps are replaced with full-one matrices without fine-tuning. Both sets of images exhibit similar visual quality.

or quantization [25, 48]. Spectral Diffusion [61] boosts structure design by incorporating frequency dynamics and priors. OMS-DPM [27] proposes a model schedule that selects small models or large models at specific sampling steps to balance generation quality and inference speed. Deepcache [33], TokenCache [29], and DiTFastAttn[66] reuse or share similar features across sampling steps. The early stopping mechanism in diffusion is explored in [24, 37, 55].

Most of the above methods are designed for general diffusion models. Recently, the Diffusion Transformer (DiT) has emerged as a more potential backbone, surpassing the previously dominant U-Net architectures. Unlike U-Net-based models, DiT's inefficiency stems largely from the self-attention mechanism. Limited research has specifically addressed the efficiency of DiT-based models. Pu et al. [42] identifies the redundancies in the self-attention and adopts timestep dynamic mediator tokens to compress attention complexity, achieving some performances but limited FLOPs reduction, as this approach primarily reduces the number of keys and values rather than the overall tokens count. In Vision transformers for classification task, numerous methods [59, 9, 15, 7, 4, 73, 36, 69, 67] successfully removed redundant tokens to improve performance-efficiency trade-offs. ToMeSD [5] is the first to explore token reduction in diffusion models. However, its performance on DiT was suboptimal, as shown in [37, 68]. U-Net can be regarded as a sparse network due to its contracting and expansive structure. EDT [10] applies a novel architecture similar to U-Net, while they still have 1.4 FID score gap compared to its baseline model MDTv2 [16] via 2,000K iterations. These findings indicate that directly transferring token reduction techniques from classification tasks or reusing U-Net structures may be insufficient for DiT. Instead, a specialized token reduction strategy is needed. DyDiT [68] and DiffCR [64] design dynamic networks to compress DiT from multiple dimensions, such as token, layer, attention head, channel.

## 3 Method

We first introduce SparseDiT architecture in Section 3.1, followed by our timestep-wise pruning rate strategy in Section 3.2. Finally, Section 3.3 details the initialization and fine-tuning of our network.

### 3.1 SparseDiT

**Overview architecture.**  The SparseDiT architecture is illustrated in Figure 2a, where the transformer layers of DiT based model are divided into three groups: bottom, middle, and top. The bottom layers leverage poolingformers to efficiently capture global features. The middle layers contain multiple sparse-dense token modules (SDTM), which decouple the representation process into global structure capture and local details enhancement using sparse and dense tokens, respectively. The top layers retain the standard transformers, processing dense tokens to generate the final predictions at each sampling step. The primary computational savings are achieved through sparse tokens, hence the majority of transformer layers are located in the middle section to maximize efficiency.

**Poolingformer in the bottom layers.**  As shown in the first subfigure in Figure 1a, attention scores in the bottom layers exhibit a nearly uniform distribution, with each token evenly extracting global features, akin to a global average pooling operator. To investigate further, we conducted a toy experiment where the attention maps in the first two transformer layers were replaced with

matrices filled with a constant value (*e.g.*, 1), without any fine-tuning. Figure 3 shows that, using the same initial noise and random seed, the images generated by the original DiT-XL and the modified DiT-XL are nearly identical, suggesting that complex self-attention calculations offer limited additional information. Given that self-attention in the bottom layers captures global features, we further question whether sparse tokens can be used in these layers. However, experiments (in Table 6) demonstrated that applying sparse tokens in the bottom layers results in unstable training, highlighting the necessity of retaining complete tokens in these layers.

Based on the above analysis, we adopt poolingformers to replace the original transformers. In poolingformer, we remove queries and keys as attention maps are not computed. Instead, we perform a global average pooling on the value $V \in \mathbb{R}^{N \times C}$ and integrate it into input tokens $X$, which can be represented as

$$X = X + \bar{V}, \tag{1}$$

where $\bar{V}$ is the mean along the dimension of $N$, *i.e.*, $\bar{v} = \frac{1}{N} \sum V (\bar{v} \in \mathbb{R}^{1 \times C})$, and then repeating $N$ times to shape $N \times C$. The parameters from adaLN block are omitted in all the equations for brevity. Our poolingformer can be viewed as a special case of the model in [65], behaving identically when the pooling kernel size is equal to the input size.

**Sparse-dense token module.** We present the sparse-dense token module (SDTM) to generate sparse and dense tokens processed by the corresponding transformer layers. The high-level idea is to decouple global structure extraction and local detail extraction. Sparse tokens capture global structure information and reduce computational cost, while dense tokens enhance local detail and stabilize training. Sparse and dense tokens are converted to each other within SDTM.

Initially, SDTM introduces a set of sparse tokens $X_s \in \mathbb{R}^{M \times C}$, where $M$ is the number of sparse tokens, typically $M \ll N$. These sparse tokens can be initialized by adaptively pooling the dense tokens $X$. We define pruning rate $r = 1 - M/N$ to represent the sparsity degree. To store the structure information, we first reshape the dense tokens $X \in \mathbb{R}^{N \times C}$ into the latent shape. For instance, if the input is an image, we reshape it into shape $H \times W \times C$, then pool it across the spatial dimensions to shape $H' \times W' \times C$, where $H' \times W' = M$. The intentions of spatial pooling initialization are two-fold. First, the initial sparse tokens can be distributed uniformly in space and the representation of each sparse token is associated with a specific spatial location, which is beneficial for downstream tasks such as image editing [20]. Second, it can prevent the semantic tokens from collapsing to one point in the following layers. Sparse token interact with full-size dense tokens via an attention layer to integrate the global information:

$$X_s = X_s + MHA(X_s, X, X), \tag{2}$$

where the triplet input of $MHA$ are queries, keys, and values in turn.

The generated sparse tokens are fed into subsequent transformer layers, referred to as sparse transformers. Since $M \ll N$, our SDTM can substantially reduce computational cost.

Following this, we restore dense tokens from sparse tokens. First, we reshape the sparse tokens $X_s \in \mathbb{R}^{M \times C}$ into structure shape $H' \times W' \times C$ and upsample it to the same shape as input dense tokens $X \in \mathbb{R}^{H \times W \times C}$. We introduce two linear layers to combine upsampling sparse tokens with dense tokens, which is represented as:

$$X_{merged} = UpSample(X_s) \cdot W_1 + X \cdot W_2, \tag{3}$$

where $W_1, W_2 \in \mathbb{R}^{C \times C}$ are the weights of two linear layers. Then, to further incorporate sparse tokens, we utilize an attention layer to perform a reverse operation of the generation of sparse token to produce the restored dense tokens, which is written as:

$$X = X_{merged} + MHA(X_{merged}, X_s, X_s). \tag{4}$$

At the end of SDTM, several transformer layers, termed dense transformers, process full-size dense tokens to enhance local details.

We cascade multiple SDTMs in our network. By repeating sparse and dense tokens, the network effectively preserves both structural and detailed information, achieving substantial reductions in computational cost while maintaining high-quality generation outputs.

Table 1: Comparison of performance between SparseDiT and DiT for class-conditional image generation task on ImageNet.

| Resolution | Model | FLOPs (G) | Throughput (img/s) | FID ↓ | sFID ↓ | IS ↑ | Precision ↑ | Recall ↑ |
|---|---|---|---|---|---|---|---|---|
| $256 \times 256$ | DiT-B | 23.01 | 7.84 | 9.07 | 5.44 | 121.07 | 0.74 | 0.54 |
|  | **Ours** ($r \in [0.61, 0.86]$) | 14.34 (-38%) | 13.2 (+68%) | 8.23 | 6.20 | 134.30 | 0.74 | 0.54 |
| $256 \times 256$ | DiT-XL | 118.64 | 1.58 | 2.27 | 4.60 | 278.24 | 0.83 | 0.57 |
|  | **Ours** ($r \in [0.44, 0.61]$) | 88.91 (-25%) | 2.13 (+35%) | 2.23 | 4.62 | 278.91 | 0.84 | 0.58 |
|  | **Ours** ($r \in [0.61, 0.86]$) | 68.05 (-43%) | 2.95 (+87%) | 2.38 | 4.82 | 276.39 | 0.82 | 0.58 |
| $512 \times 512$ | DiT-XL | 525 | 0.249 | 3.04 | 5.02 | 240.82 | 0.84 | 0.54 |
|  | **Ours** ($r \in [0.61, 0.86]$) | 286 (-46%) | 0.609 (+145%) | 2.96 | 5.00 | 242.4 | 0.84 | 0.54 |
|  | **Ours** ($r \in [0.90, 0.96]$) | 235 (-55%) | 0.685 (+175%) | 3.13 | 5.41 | 236.56 | 0.83 | 0.52 |

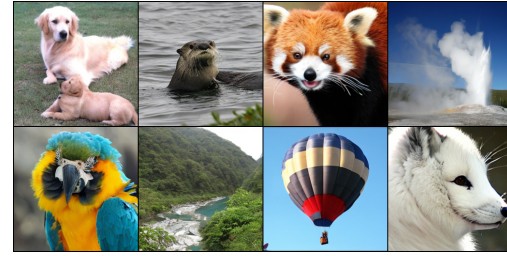 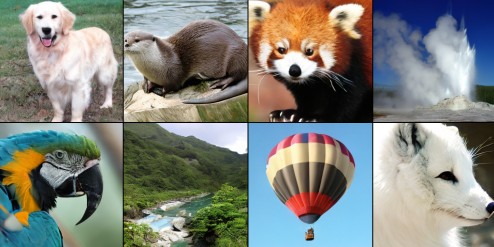

(a) Images generated by DiT.

(b) Images generated by SparseDiT (-43% FLOPs).

Figure 4: Generating images from DiT-XL and SparseDiT-XL with the same random seed.

## 3.2 Timestep-wise pruning rate

We have observed that the tokens display varying denoising behavior over sampling steps. They generate the low-frequency global structure information in the early denoising stage and the high frequency details in the late denoising stage. The token count requirement is progressively increase along sampling steps. We exploit this by dynamically adjusting the pruning rate $r$ across sampling steps, increasing tokens as sampling progresses.

Given $T$ sampling steps, we propose a sample-specific approach to dynamically adjust the pruning rate $r$, thereby controlling the number of sparse tokens across the sampling steps. We define a range for $r$, such that $r \in [r_{\min}, r_{\max}]$. Observing that generation quality is highly relative with the later denoising stages (in Section 4.4), we hold $r$ constant at $r_{\min}$ for the first $T/4$ sampling steps. Subsequently, we adjust the pruning rate $r$ linearly based on the current sampling step $t_i$. The specific formula for $r$ is provided as follows:

$$r = \begin{cases} r_{min}, & t_i < T/4, \\ \frac{4T-4}{3T}r_{min} + \frac{4-T}{3T}r_{max}, & T/4 \le t_i < T. \end{cases} \quad (5)$$

During training, we sample step $t_i$ and compute the corresponding $r$ according to Eq. 5. However, the input tokens should be the same to train the model in batch, and the randomness of the sampling from $T$ should also be maintained. To solve this contradiction, we modify the linear function of Eq. 5 when $T/4 \le t_i < T$ to a piecewise function. Normally, the model is trained by multiple GPUs. We request $t_i$ sampled in a specific piece in each GPU. Therefore, in each iteration, it can achieve both uniformly random sampling and batch training.

## 3.3 Initialization and fine-tuning

Our method fine-tunes pre-trained DiT models to improve efficiency. The processes of generating sparse tokens and restoring dense tokens utilize off-the-shelf transformers, avoiding the need for additional networks. During fine-tuning, the parameters of transformers in DiT-based model are loaded into the corresponding transformers in our SparseDiT, with two exceptions: (1) queries and keys are absent in Poolingformers, hence related parameters in pre-trained model are not required; (2) the weights $W_1$ and $W_2$ in Eq. 3 are initialized by full zeros and identity matrix, receptively.

Table 2: FVD scores of SparseDiT-XL in class-conditional video generation task on four mainstream video generation datasets. We do not find the official reference sets of these 4 dataset when we compute FVD scores, hence we build the reference sets by ourselves and use the official checkpoint to re-compute the FVD scores of Latte-XL in this table.

| Model | FLOPs (G) | Throughput (clip/s) | FaceForensic | SkyTimelapse | UCF101 | Taichi-HD |
|---|---|---|---|---|---|---|
| Latte-XL* | 1894 | 0.108 | 24.10 | 40.78 | 284.54 | 89.63 |
| **Ours** ($r \in [0.80, 0.93]$) | 828 (-56%) | 0.228 (+111%) | 24.51 | 39.02 | 288.90 | 84.13 |

# 4 Experiments

SparseDiT is applied in three representative DiT-based models, DiT [41], Latte [34], and PixArt-$\alpha$ [8] for class-conditional image generation, class-conditional video generation, and text-to-image generation, respectively.

**Fine-tuning overhead.** All training settings and hyperparameters follow their respective papers. Fine-tuning requires approximately 6% of the time needed for training from scratch, *e.g.*, 400K iterations for DiT-XL fine-tuning.

## 4.1 Class-conditional image generation

**Experimental setting.** We conduct our experiments on ImageNet-1k [11] at resolutions of $256 \times 256$ and $512 \times 512$, following the protocol established in DiT. For DiT-XL, the model consists of 2, 24, and 2 transformers in the bottom, middle, and top segments, respectively. The middle segment includes 4 sparse-dense token modules, which comprise 1 sparse token generation transformer, 3 sparse transformers, 1 dense token recovery transformer, and 1 dense transformer. For DiT-B, the bottom, middle, and top segments contain 1, 10, and 1 transformer layers, respectively. The middle segment consists of 2 sparse-dense token modules, including 1 sparse token generation transformer, 2 sparse transformers, 1 dense token recovery transformer, and 1 dense transformer. As poolingformers and sparse transformers can interfere with the function of position embedding, we reintroduce sine-cosine position embedding at each stage of dense token generation. For DiT-XL, we use the checkpoint from the official DiT repository, while for DiT-B, we utilize the checkpoint provided by [40]. Following prior works, we sample 50,000 images to compute the Fréchet Inception Distance (FID) [17] using the ADM TensorFlow evaluation suite [12], along with the Inception Score (IS) [46], sFID [38], and Precision-Recall metrics [23]. Classifier-free guidance [19] (CFG) is set to 1.5 for evaluation and 4.0 for visualization. Throughput is evaluated with a batch size of 128 on an Nvidia A100 GPU.

**Main results.** The results of our approach on the class-conditional image generation task using ImageNet are presented in Table 1. We evaluate our method on two model sizes, DiT-B and DiT-XL, and at two resolutions, $256 \times 256$ and $512 \times 512$. At pruning rates $r \in [0.61, 0.86]$, SparseDiT-XL achieves a 43% reduction in FLOPs and an 87% improvement in inference speed, with only a 0.11 increase in FID score. This indicates significant redundancy within DiT, as using only about 25% tokens in specific layers maintains similar performance levels. By increasing the token count, we observe slight improvements over baseline performance with 145% improvement in throughput. The impact of our approach is even more pronounced at a resolution of $512 \times 512$. With a higher pruning rate than the $256 \times 256$ resolution, our method yields a superior performance-efficiency trade-off. By pruning over 90% of the tokens, we achieve a 55% reduction in FLOPs and a 175% increase in throughput, with only a 0.09 increase in FID score. When reducing the pruning rate further, our results slightly surpass the baseline, delivering a 145% throughput improvement. ToMeSD [5] is the first approach to reduce tokens. However, it applies token merging and recovery at every layer in a generic manner, resulting in a significant performance drop. For example, with a 0.1 merging rate on DiT-XL, ToMeSD achieves an FID score of 14.74, which is substantially worse than our result. A detailed comparison between other methods and our method is provided in Appendix A.1.

From $256 \times 256$ to $512 \times 512$, FLOPs increase by only 4.4 times, whereas the speed declines by a factor of 6.3, demonstrating that the primary speed bottleneck in DiT architecture is the number of tokens. Our method effectively reduces token count, resulting in significant gains, particularly for high-resolution content generation. For example, we achieve a 55% reduction in FLOPs at $512 \times 512$ while enhancing speed by 175%. Compared to other methods, our real-world speed improvements far exceed those achieved with similar reductions in FLOPs.

Table 3: Text-to-image generation on SAM dataset. $r \in [0.61, 0.86]$.

| Model | FLOPs (G) | Throughput(img/s) | FID $\downarrow$ |
|---|---|---|---|
| PixArt-$\alpha$ | 148.73 | 0.414 | 4.53 |
| Ours | 91.62 (-38%) | 0.701 (+69%) | 4.29 |

**Visualization results.** To further validate the effectiveness of our method, we visualize some samples generated from SparseDiT-XL at a $256 \times 256$ resolution with a pruning rate $r \in [0.61, 0.86]$ (-43% FLOPs) and compare them with DiT-XL in Figure 4. Each cell in the same position corresponds to images generated from the same random seed and class. Since our model is fine-tuned from the pre-trained DiT, the overall styles and structures of the two subfigures are similar. The images generated by SparseDiT still retain rich high-frequency details. Furthermore, SparseDiT images demonstrate more precise structure, as evidenced by the accurate depiction of the "golden retriever's" nose and the "macaw's" eyes, which appear misplacement or missing in the images generated by DiT-XL. We provide samples at a resolution of $512 \times 512$ in Appendix.

## 4.2 Class-conditional video generation

**Experimental settings.** We conduct experiments at a resolution of $256 \times 256$ on four public datasets: FaceForensics [45], SkyTimelapse [60], UCF101 [54], and Taichi-HD [50]. Latte comprises two distinct types of transformer blocks: spatial transformers, which focus on capturing spatial information, and temporal transformers, which capture temporal information. To accommodate this separation of spatial and temporal feature extraction, we adjust our model's schedule within the sparse-dense token module. Specifically, we employ two transformers to prune spatial tokens and temporal tokens separately. The sparse tokens are then processed by two sparse transformers, and finally two transformers are used to recover the temporal and spatial tokens. Dense transformers are discarded in this setup. All other model configurations are the same as in DiT, and we follow the original training settings and hyperparameters used for Latte. To evaluate performance, we sample 2,048 videos, each consisting of 16 frames, and measure the Fréchet Video Distance (FVD) [57]. Throughput is measured using a batch size of 2 clips on an Nvidia A100 GPU.

**Main results.** Table 2 presents the main results of our approach. Leveraging the additional temporal dimension in video data allows for a higher pruning rate. Our method achieves a 56% reduction in FLOPs while maintaining a competitive FVD score compared to the baseline, demonstrating its effectiveness in video generation tasks.

## 4.3 Text-to-image generation

**Experiment setting.** We further assess the effectiveness of our method on text-to-image generation, which presents a greater challenge compared to class-conditional image generation. We adopt PixArt-$\alpha$ [8], a text-to-image generation model built based on DiT as the base model. Our model is initialized using the official PixArt-$\alpha$ checkpoint pre-trained on SAM dataset [22] containing 10M images. We further fine-tune it with our method on a subset of SAM dataset including 1M images. In PixArt-$\alpha$, the transformer architecture includes two types of attention layers: a self-attention layer and a cross-attention layer, which integrates textual information. We apply our method to the self-attention layers to either reduce or recover tokens.

For evaluation, we randomly select text prompts from SAM dataset and adopt 100-step IDDPM solver [39] to sample 30,000 images, with the FID score serving as the evaluation metric. Throughput is evaluated with a batch size of 128 on an Nvidia A100 GPU.

**Main results.** As shown in Table 3, our SparseDiT achieves an FID score comparable to the original PixArT-$\alpha$ with significantly accelerating the generation, showing that our method is effective for text-to-image generation task.

**Visualization results.** We visualize some samples of our method and compare them with original PixArt-$\alpha$ in Figure 5. The classifier-free guidance is set to $4.5$. The results demonstrate that our method effectively maintains both image quality and semantic fidelity.

## 4.4 Ablation study

All the following ablation experiments of SparseDiT-XL are conducted on ImageNet at a $256 \times 256$ resolution.

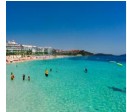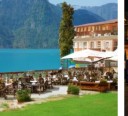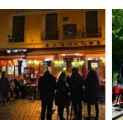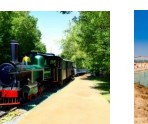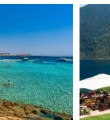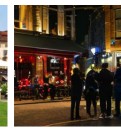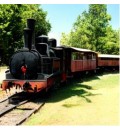

(a) Images generated by PixArt-$\alpha$        (b) Images generated by SparseDiT

Figure 5: Generating images from PixArt-$\alpha$ (a) and SparseDiT (b). The captions in the same positions are identical. They are provided in Appendix A.3.

Table 4: Performance evaluation on different numbers of SDTMs. "NAN" indicates loss nan.

| No. of SDTMs | 1 | 2 | 3 | 4 |
|---|---|---|---|---|
| FID ↓ | NAN | 3.86 | 2.51 | 2.38 |
| FLOPs | 68.74 | 67.00 | 69.97 | 68.05 |

Table 5: Combination our method with efficient samplers. The numbers represent FID scores.

| Model | 250-DDPM | 25-DDIM | 5-RFlow |
|---|---|---|---|
| DiT-XL | 2.27 | 2.89 | 43.40 |
| Ours | 2.38 | 3.31 | 43.69 |

**The number of SDTMs.** The alternation between sparse and dense tokens is a key factor contributing to the success of our method. Sparse tokens capture global structures, while dense tokens capture detailed information. In SparseDiT, we adopt four SDTMs by default. Table 4 shows the effect of using different numbers of SDTMs. To isolate the impact of SDTM count, we maintain the same FLOPs across models by adjusting the number of sparse and dense transformers while varying only the number of SDTMs. In configurations with only one SDTM, the model is modified into a U-shaped architecture. The results indicate that fewer SDTMs reduce interactions between global and local information, resulting in weaker performance. Furthermore, reducing SparseDiT to a U-Net structure compromises training stability, leading to a collapse. We do not experiment with more than four SDTMs, as increasing SDTM count beyond four offers diminishing returns in FLOP reduction.

**Combination with efficient samplers.** Our SparseDiT is a model compression method that can be seamlessly integrated with efficient samplers, such as DDIM [51] and Rectified Flow (RFlow) [28]. Note that the RFlow variant of DiT was trained by us via 2,000K iterations. As shown in Table 5, when our method is combined with the 25-step DDIM and 5-step Rectified Flow samplers, it achieves approximately $18.7\times$ and $93.4\times$ improvements in inference speed, respectively, compared to the standard 250-step DDPM. Notably, our method does not introduce a significant performance gap relative to the baseline, demonstrating that it can be effectively combined with efficient samplers.

**The number of poolingformers.** In Table 6, we examine the impact of varying the number of poolingformers in the model. We keep the total number of transformers in the bottom and top segments constant across configurations. When using three poolingformers, performance declines substantially due to the global average pooling operator's limited ability to adaptively capture information. For configurations with one or two poolingformers, the models achieve same results. The poolingformer is more efficient and consume fewer parameters than the standard transformer; therefore, we default to using 2 poolingformers. Reducing the poolingformer count to zero results in training instability, suggesting that maintaining full-size tokens in the initial layers is crucial. This finding aligns with the observations in Figure 1b: in DiT, the first two attention maps exhibit a uniform distribution, allowing us to replace them with poolingformers, but further replacement disrupts information flow and reduces performance.

**The effectiveness of timestep-wise pruning rate.** To evaluate the effectiveness of the timestep-wise pruning rate strategy, we conduct ablation experiments, as shown in Table 7. In these experiments, we maintain the same number of FLOPs while varying the number of tokens along timesteps, either using a constant token count or dynamically adjusting the token count. The results demonstrate that dynamically tuning the pruning rate significantly improves performance. Additionally, Table 7 highlights an observation: the performance of the dynamic pruning strategy is particularly relative

Table 6: Performance evaluation on different numbers of poolingformers. "NAN" indicates loss nan during training.

| No. of Poolingformers | 0 | 1 | 2 | 3 |
|---|---|---|---|---|
| No. of trans in top seg. | 4 | 3 | 2 | 1 |
| FID ↓ | NAN | 2.38 | 2.38 | 2.56 |
| FLOPs | 69.56 | 68.80 | 68.05 | 67.29 |

Table 7: Comparison of FID scores with and without timestep-wise pruning rate strategy.

| FLOPs (G) | No. of Tokens | Dynamic | FID ↓ |
|---|---|---|---|
| ∼ 68 | $8 \times 8$ | | 2.48 |
| | $6 \times 6 \sim 10 \times 10$ | ✓ | 2.38 |
| ∼ 61 | $6 \times 6$ | | 2.63 |
| | $4 \times 4 \sim 8 \times 8$ | ✓ | 2.50 |

with the token count of the later stages of denoising. For instance, the configuration with token counts ranging from $4 \times 4$ to $8 \times 8$ only shows a 0.02 FID score increase compared to the configuration with a constant $8 \times 8$ token count.

## 5 Conclusion

**Limitations.** The primary limitation of our method arises from the manually pre-defined its structure, containing the number of layers in each module and the number of sparse tokens.

In this paper, we introduce SparseDiT, a novel approach to enhancing the efficiency of DiT by leveraging token sparsity. By incorporating SDTM and a timestep-wise pruning rate in a strategically layered manner, SparseDiT significantly reduces computational complexity while preserving high-generation quality. Experimental results show substantial FLOP reductions with minimal performance degradation, demonstrating SparseDiT's effectiveness across multiple DiT-based models and datasets. This work marks a step forward in developing scalable and efficient diffusion models for high-resolution content generation, expanding the practical applications of DiT-based architectures.

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

# A   Appendix

## A.1   Comparison with other methods

**Comparison with ToMeSD.**   Recent studies [37, 68] have evaluated ToMeSD [5] on DiT models and reported significant performance drops, which are much lower than those achieved by our method. Our approach is fundamentally different from ToMeSD. While ToMeSD reduces and recovers tokens generically at every layer, our method specifically analyzes the local-global relationships unique to DiT models. Based on these insights, we strategically reduce and recover tokens. We have evaluated ToMeSD on DiT and compared it with our method, as shown in Table 8.

**Comparison with DyDiT.**   DyDiT [68] represents the current state-of-the-art approach for enhancing the efficiency of Diffusion Transformers (DiTs). In contrast to our method, which emphasizes token reduction, DyDiT implements pruning across tokens, attention heads, and channels. A comparative analysis of our method against DyDiT, as detailed in Table 9 using ImageNet 512×512 images, illustrates a substantial advantage in computational efficiency for our approach. Specifically, our method achieves a 46% reduction in FLOPs, markedly surpassing DyDiT's 29% reduction, coupled with a 145% increase in inference speed, significantly outperforming the 31% improvement achieved by DyDiT. The only trade-off is a minor increase in the Fréchet Inception Distance (FID), where our method's FID is 0.08 higher than that of DyDiT. Nevertheless, this difference in FID is nearly imperceptible to human observers in practical applications.

**Comparison with TokenCache.**   We compare our method with TokenCache [29] in Table 9. We adopt the standard DiT baseline 2.27, while TokenCache applies a baseline 2.24. TokenCache achieves a 32% speed gain with a slight FID increase. Our approach attains an 87% speed gain with a smaller FID increase, demonstrating superior efficiency and performance.

**Comparison with Ditfastattn.**   We compare our method with Ditfastattn [66] in Table 9. We adopt the standard DiT baseline 3.04, while Ditfastattn applies a baseline 3.16. Ditfastattn shows a substantial FID increase for speed gain, while our method results in better speed and less FID.

## A.2   Training form scratch

We train our SparseDiT from scratch, reaching 400K training iterations. We compare our result with original DiT paper in Table 4. The results (cfg=1.0) are in Table 10.

## A.3   Captions in Figure 5

Column 1: The image depicts a beach scene with a large body of water, such as a lake or ocean, and a sandy shoreline. The beach is filled with people, including a group of people swimming in the water.

Column 2: The image depicts a beautiful outdoor dining area with a large number of tables and chairs arranged in a row, overlooking a picturesque lake. The tables are covered with white tablecloths, and there are several umbrellas providing shade for the guests. The scene is set in a lush green field, with a large building in the background, possibly a hotel or a restaurant. The tables are arranged in a way that allows for an unobstructed view of the lake, creating a serene and relaxing atmosphere for the diners. The image has a stylish and elegant feel, with the attention to detail in the table arrangement and the choice of location contributing to a memorable dining experience.

Column 3: The image depicts a busy city street at night, with a group of people standing outside a restaurant and a bar. The scene is set in a European city, and the atmosphere is lively and bustling.

Column 4: The image features a vintage-style train, parked on a track surrounded by trees and grass. The train appears to be an old-fashioned steam engine, which is a type of locomotive powered by steam. The train is positioned in a park-like setting, with a tree-lined path nearby. The scene is set in a sunny day, creating a pleasant atmosphere. The image has a nostalgic and historical feel, evoking a sense of the past and the charm of old-time trains.

Table 8: Comparison of our method with ToMeSD on DiT models.

| Model | Method | FID | Speed-up |
|-------|--------|-----|----------|
| DiT-B | ToMeSD | 29.24 | +20% |
|       | Ours | 8.23 | +68% |
| DiT-XL | ToMeSD | 14.74 | +66% |
|        | Ours | 2.38 | +87% |

Table 9: Comparison of our method with DyDiT on DiT models.

| Model | Resolution | Method | FLOPs | FID | IS | Speed-up |
|-------|-----------|--------|-------|-----|-----|----------|
| DiT-XL | $512 \times 512$ | DyDiT | 375 (-29%) | 2.88(-0.16) | - | +31% |
|        |           | Ditfastattn | - | 4.52(+1.36) | 180.34 | +98% |
|        |           | Ours | 286 (-46%) | 2.96(-0.08) | 242.4 | +145% |
| DiT-XL | $256 \times 256$ | TokenCache | 72.25(-39%) | 2.37 (+0.13) | 262.00 | +32% |
|        |           | Ours | 68.05(-43%) | 2.38 (+0.11) | 276.39 | +87% |

## A.4 Additional visualization

We provide additional visualizations at a resolution of $512 \times 512$ on SparseDiT-XL from Figure 6 to Figure 17. The class labels, including "arctic wolf", "volcano", "cliff drop-off", "balloon", "sulphur-crested cockatoo", "lion", "otter", "coral reef", "macaw", "red panda", ""husky", and "panda", correspond to the same cases presented in DiT. Readers can compare our results with those of DiT. Our samples demonstrate comparable image quality and fidelity. The classifier-free guidance scale is set to 4.0, and all samples shown here are uncurated.

Table 10: Comparison of our method with ToMeSD on DiT models.

| Method | FID | Flops | Interations |
|--------|-----|-------|-------------|
| DiT-XL | 19.47 | 118.64 | 400K |
| Ours (from scratch) | 15.11 | 68.05 | 400K |

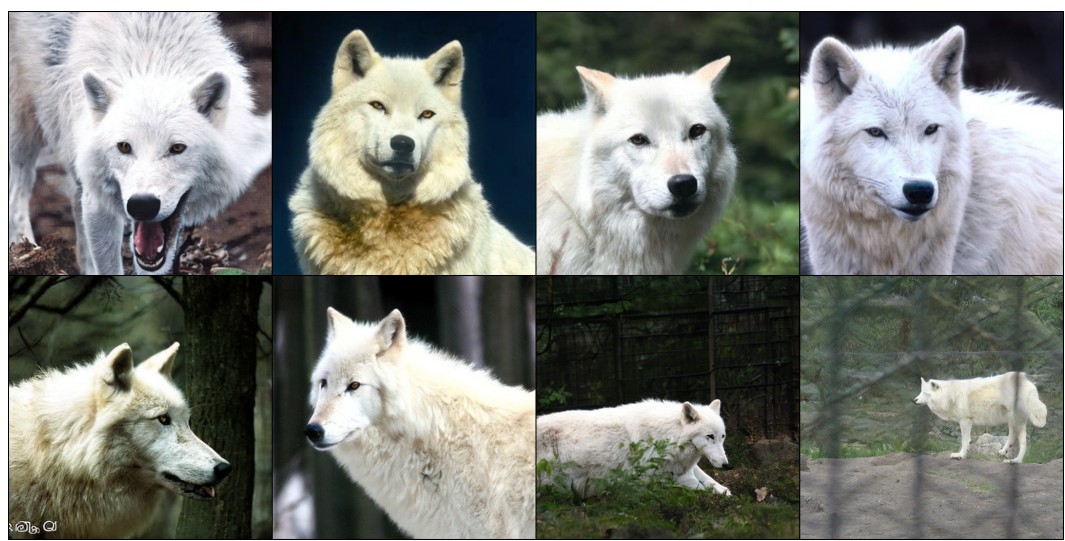

Figure 6: Uncurated $512 \times 512$ SparseDiT-XL samples.
Classifier-free guidance scale = 4.0
Class label = "arctic wolf" (270)

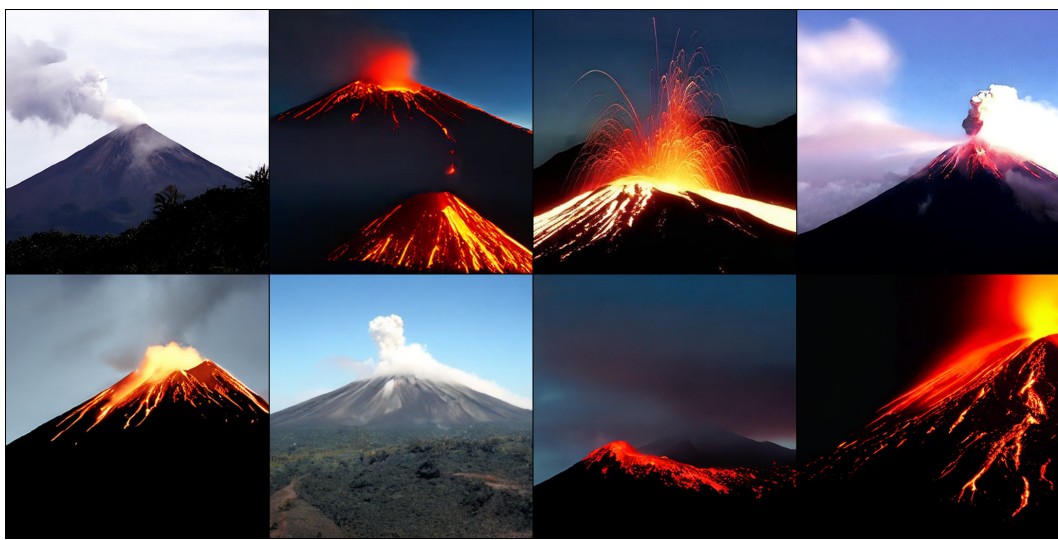

Figure 7: Uncurated $512 \times 512$ SparseDiT-XL samples.
Classifier-free guidance scale = 4.0
Class label = "volcano" (980)

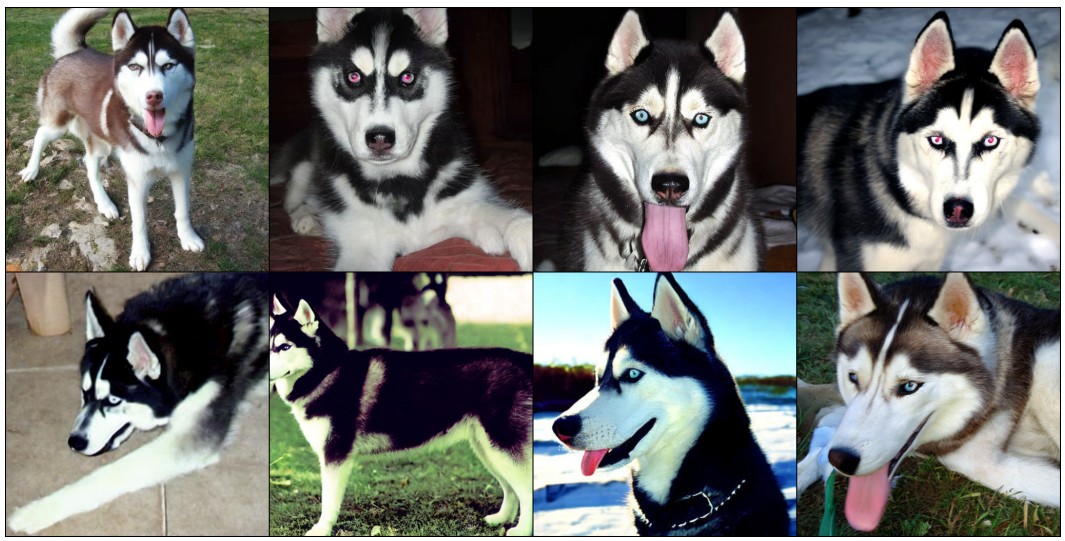

Figure 8: Uncurated $512 \times 512$ SparseDiT-XL samples.
Classifier-free guidance scale = 4.0
Class label = "husky" (250)

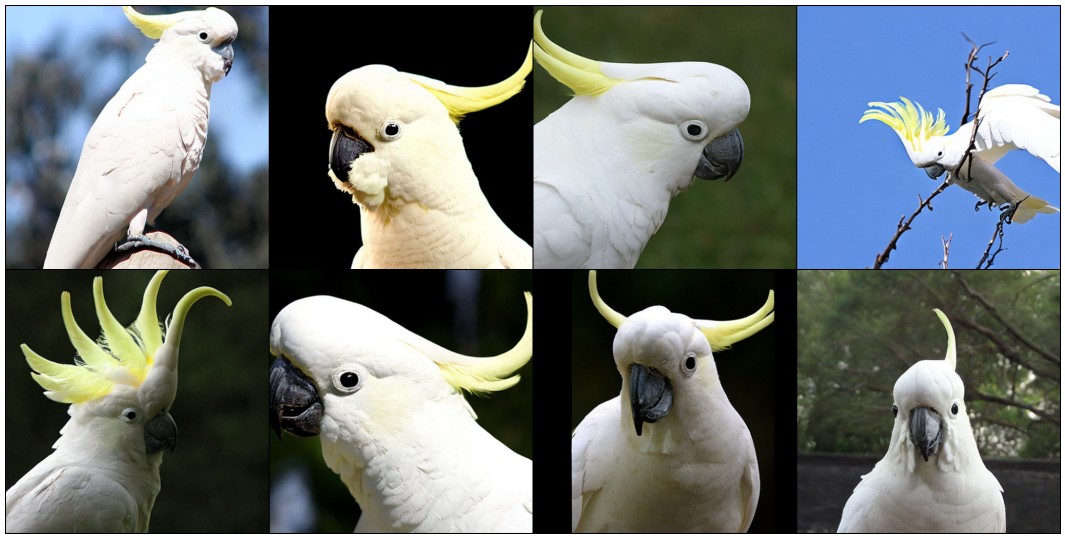

Figure 9: Uncurated $512 \times 512$ SparseDiT-XL samples.
Classifier-free guidance scale = 4.0
Class label = "sulphur-crested cockatoo" (89)

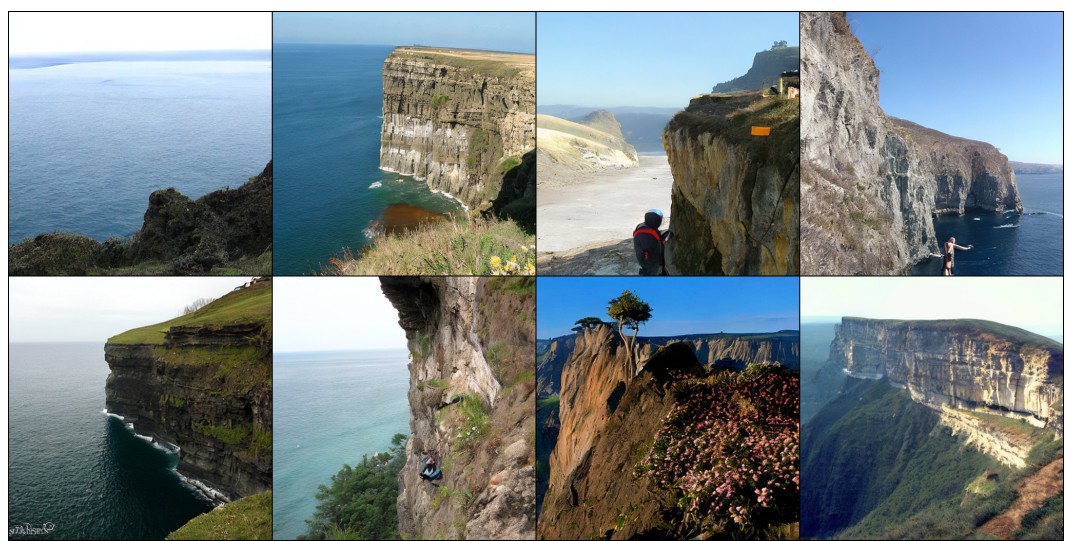

Figure 10: Uncurated $512 \times 512$ SparseDiT-XL samples.
Classifier-free guidance scale = 4.0
Class label = "cliff drop-off" (972)

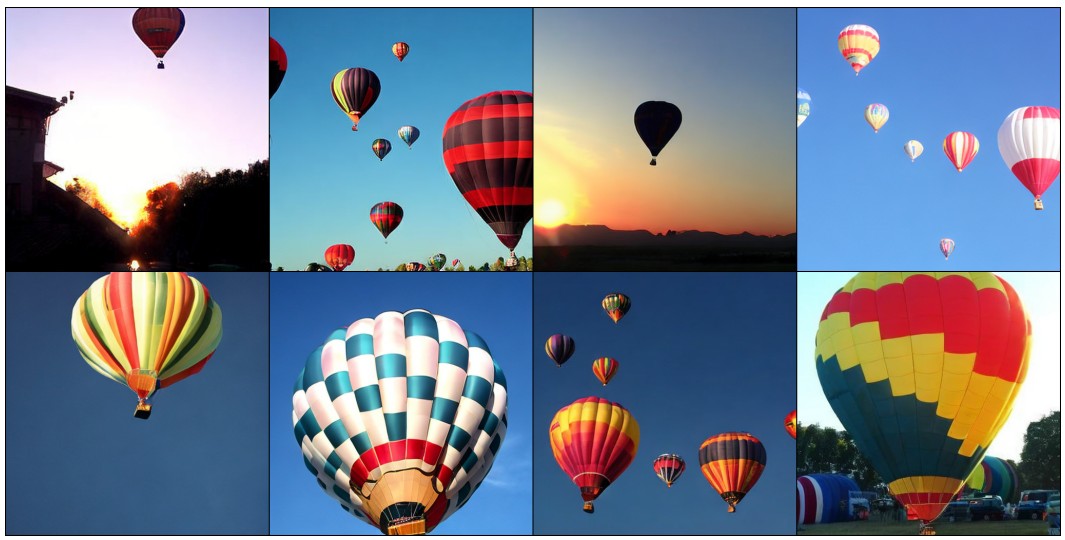

Figure 11: Uncurated $512 \times 512$ SparseDiT-XL samples.
Classifier-free guidance scale = 4.0
Class label = "balloon" (417)

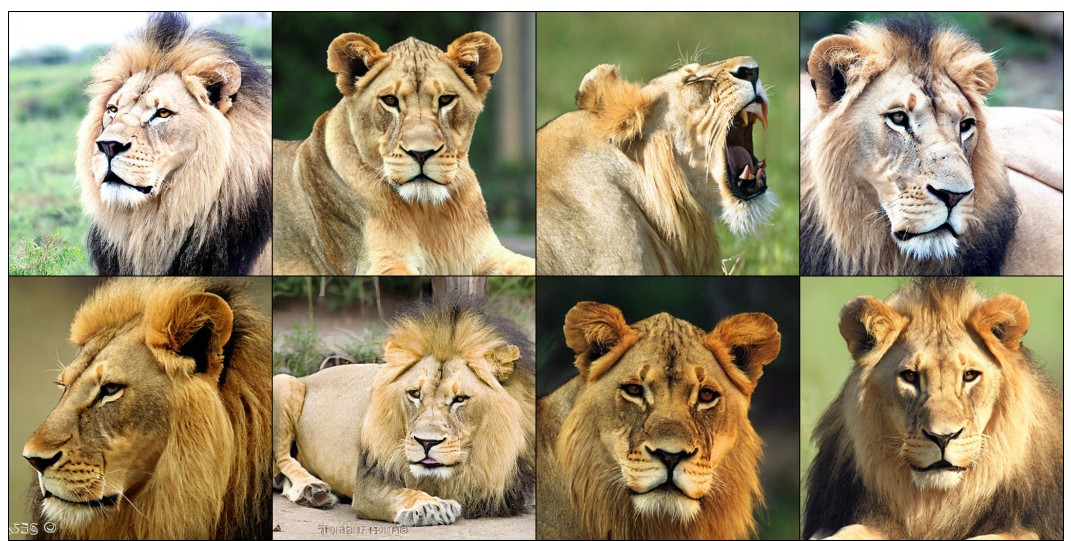

Figure 12: Uncurated $512 \times 512$ SparseDiT-XL samples.
Classifier-free guidance scale = 4.0
Class label = "lion" (291)

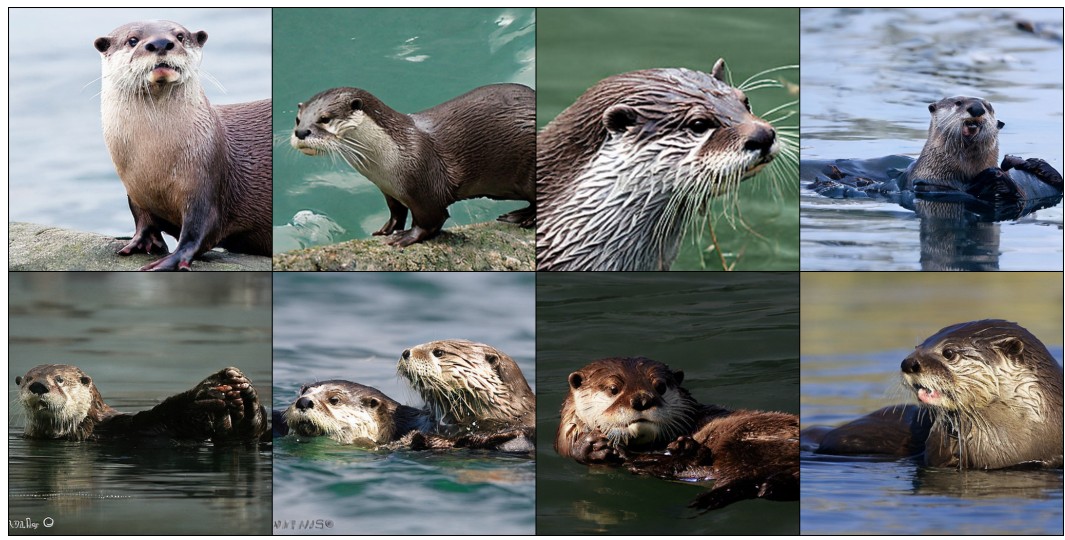

Figure 13: Uncurated $512 \times 512$ SparseDiT-XL samples.
Classifier-free guidance scale = 4.0
Class label = "otter" (360)

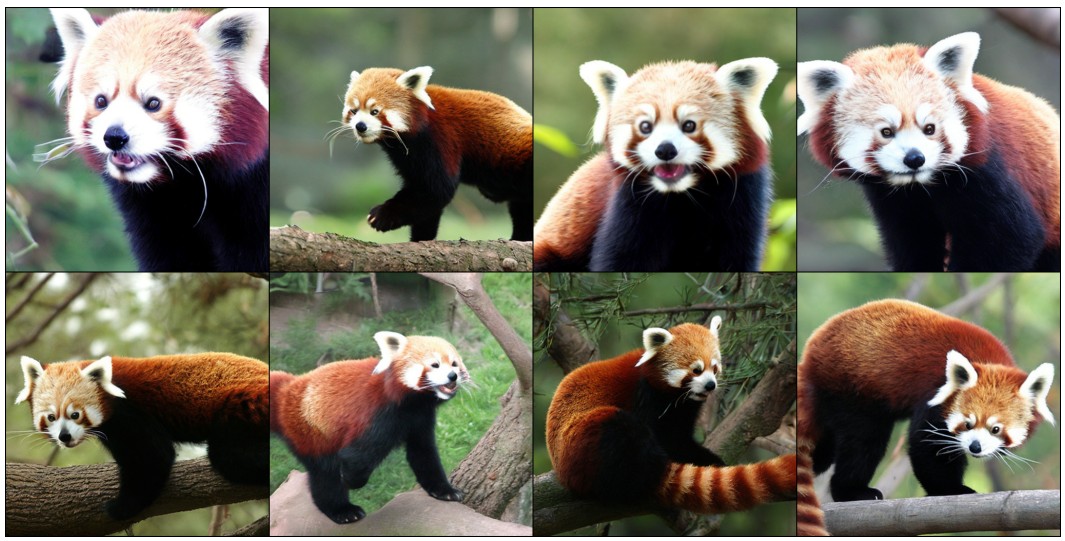

Figure 14: Uncurated $512 \times 512$ SparseDiT-XL samples.
Classifier-free guidance scale = 4.0
Class label = "red panda" (387)

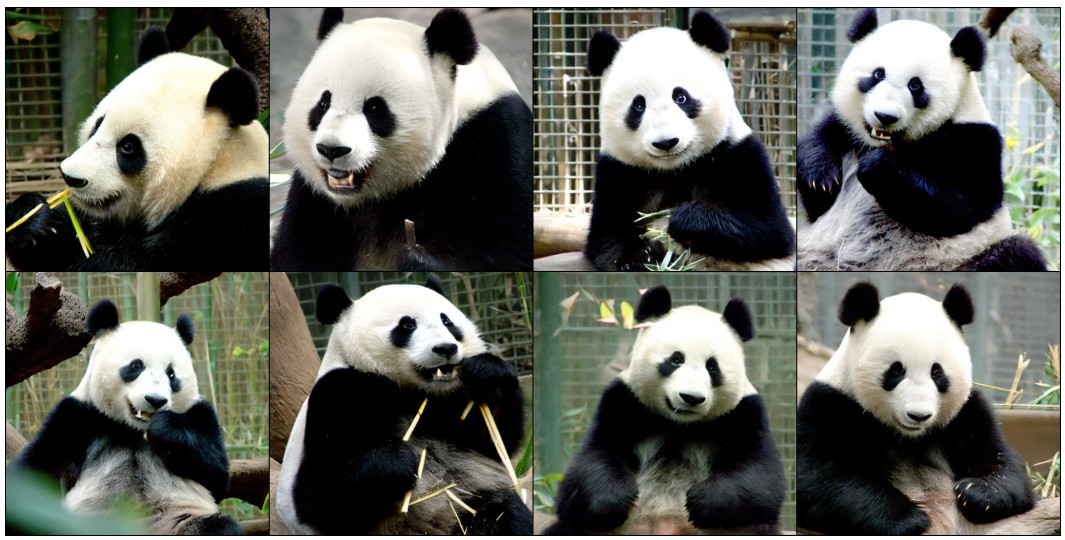

Figure 15: Uncurated $512 \times 512$ SparseDiT-XL samples.
Classifier-free guidance scale = 4.0
Class label = "panda" (388)

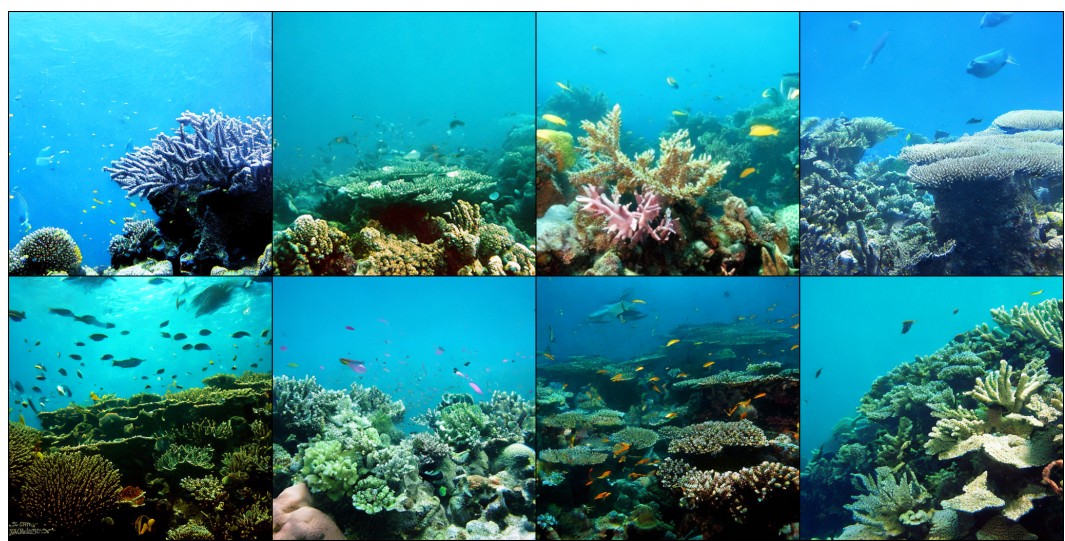

Figure 16: Uncurated $512 \times 512$ SparseDiT-XL samples.
Classifier-free guidance scale = 4.0
Class label = "coral reef" (973)

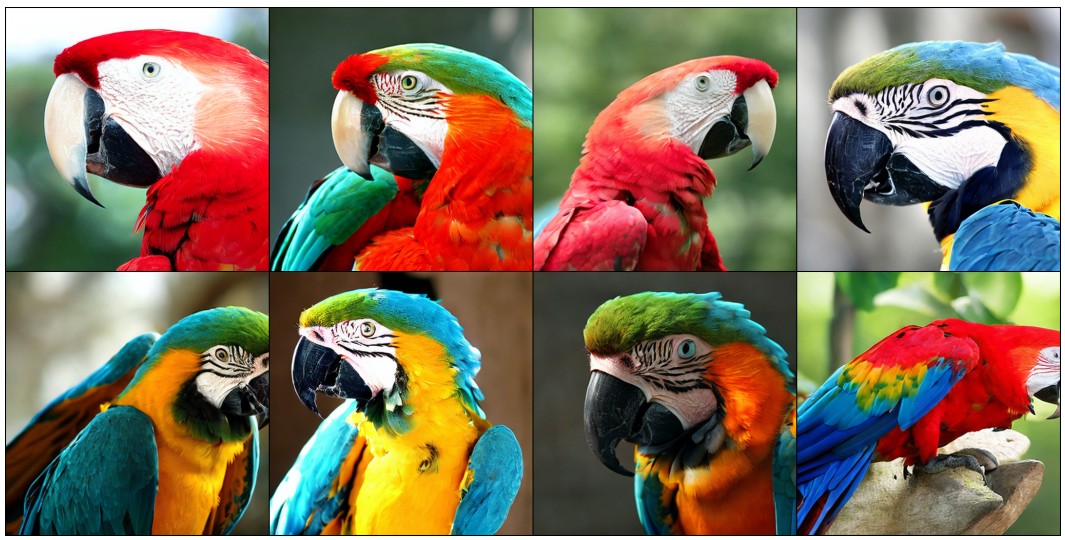

Figure 17: Uncurated $512 \times 512$ SparseDiT-XL samples.
Classifier-free guidance scale = 4.0
Class label = "macaw" (88)

