# OpenReview forum: "SparseDiT: Token Sparsification for Efficient Diffusion Transformer"
_NeurIPS.cc/2025/Conference — NeurIPS 2025 poster_

### Official Review · Reviewer_rhnM · 2025-06-27

**Clarity:** 2
**Significance:** 3
**Originality:** 3
**Rating:** 5
**Confidence:** 4

**Summary:**

The paper proposes SparseDiT, a new architecture that avoids redundant computations in diffusion transformers, leading to increased throughput at minimal or no quality cost. Compared to the normal DiT, the authors introduce two components. First, poolingformer, which simply removes the computation from queries and keys, and simply average pools and repeats the values of the first layer's attention to save the quadratic attention computation (in the first few transformer blocks). And second, sparse-dense token modules (SDTM), which cross-attend to a pooled version of the dense tokens, perform attention with sparse tokens, and then upsample the tokens back to dense tokens. The authors propose a timestep-wise pruning strategy for the number of tokens in the SDTM modules to achieve a balance between performance and speed. The authors show the superiority of their method on class-conditional and text-conditional image generation, as well as video generation.

**Questions:**

I believe the proposed method has clear potential. However, to consider increasing my final rating, I would appreciate clarification on the following key points (which were already mentioned above):

1. Clarification on baselines and direct empirical comparison to other test-time caching or attention compression methods.
2. Direct comparison of FLOPs, parameters, and throughput for a DiT block versus SDTM, to isolate the efficiency gains due to token sparsity.
3. Isolate the gain from the proposed architectural components compared to continued training.

**Ethical Concerns:**

["NO or VERY MINOR ethics concerns only"]

**Final Justification:**

The authors clarified all my remaining questions in the rebuttal.

**Limitations:**

The limitations section mainly consists of a single sentence that describes the main limitation of requiring manual analysis of the structure design. This neither discusses the limitation nor provides any explanation or solution to this problem. Hence, I would consider this as insufficient.

**Quality:**

2

**Strengths And Weaknesses:**

**Strengths**

The paper is well-organized and well-written, with a clear common thread. A key strength lies in the comprehensive experimental evaluation, which spans class-conditional and text-conditional image generation as well as video synthesis, demonstrating the broad applicability. The proposed modules are simple, yet effective in increasing the throughput of diffusion transformers. The spatially-adaptive design based on their empirical analysis of attention is interesting. Additionally, the timestep-aware pruning strategy is both intuitive and effective. Overall, I think it is a solid and simple method to improve efficiency.

**Weaknesses**

- Lack of baselines: While the paper provides many internal comparisons, it lacks comparisons to other strong baseline methods. In the supplementary material, the authors compare to DyDiT and ToMeSD, however, there exist other test-time caching methods, some of which are mentioned in the related works section, but not empirically compared to (i.e., [1, 2, 3]). I would consider direct performance comparisons to be crucial since some of these methods avoid computing tokens entirely and do not require retraining the base model at all. The authors also do not compare their method to attention compression methods (e.g., [4]).
- Potential mischaracterization of flow-based methods: The paper describes flow-based approaches (e.g., Rectified Flow) as methods that “straighten transport trajectories” (l.99), which may oversimplify their role. In practice, Rectified Flow provides a more general framework that modifies aspects like the interpolant and training target [5], rather than explicitly straightening trajectories. While trajectory straightening can be achieved through techniques like ReFlow [2], which trains on image-noise pairs from a pre-trained flow model, this is not a property of Rectified Flow per se (albeit found in some papers empirically, e.g. [5, 7]). This conceptual ambiguity also makes the comparison between DDIM and RFlow in Table 5 somewhat unclear. Since DDIM is closely related to Euler discretization in Rectified Flow, a more representative comparison of samplers would involve solvers such as DPM [8] or DPM++ [9].
- Lack of analysis protocol: The proposed tri-segment design and pruning strategy requires prior analysis on a trained model. Hence, the method can be considered as a post-training strategy. However, the analysis on which the authors ground their method doesn't establish a clear and systematic protocol for selecting key hyperparameters (e.g. number of SDTM modules). While the justification for using poolingformer layers is reasonable, it remains unclear how the authors infer the distinction between local and global attention solely from the normalized attention variances across layers. A more systematic protocol and detailed analysis would definitely strengthen the paper.
- Missing isolated analysis of SDTM vs DiT block: The SDTM module introduces additional parameters and FLOPs through the two cross-attention operations and the two linear mixing matrices $W_1, W_2 \in \mathbb{R}^{C \times C}$. I would highly appreciate a direct comparison of throughput, FLOPs, and parameter counts between a standard DiT block and SDTM. This would clarify whether the efficiency gains are truly due to token sparsity.
- Confound metrics from finetuning: The improved FID scores reported e.g., in table 1, could also stem from continued training of the DiT rather from the proposed architectural changes. A comparison to the DiT-XL and PixArt-$\alpha$ baselines with continued training would help isolate the effect of the proposed method.

**General Questions and Clarifications**
- Usually DiT models contain high norm tokens. Did you observe similar patterns in your analysis?
- What kind of token upsampling do you use for your method? How do you handle different sizes for floating point numbers of $r$?
- l266ff: You mention that SparseSiT gives a more precise structure. Is this consistent behaviour or are these just artefacts from the initial noise? For example, sometimes one seed might work better for your method than for the baseline. Can you provide any support for this claim or an intuitive explanation?

**Minors**
- writing (L96ff and L193ff)
- improved readability of the settings for DiT-B and DiT-XL in experimental section (e.g. in form of a table)
- label of table 5 and table 7 seem to be switched
- lack of other metrics for the T2I setting (e.g. CLIP score, etc)


**References**
- [1] Wimbauer, F., Wu, B., Schoenfeld, E., Dai, X., Hou, J., He, Z., ... & Wang, J. (2024). Cache me if you can: Accelerating diffusion models through block caching. In Proceedings of the IEEE/CVF Conference on Computer Vision and Pattern Recognition (pp. 6211-6220).
- [2] Ma, X., Fang, G., & Wang, X. (2024). Deepcache: Accelerating diffusion models for free. In Proceedings of the IEEE/CVF conference on computer vision and pattern recognition (pp. 15762-15772).
- [3] Lou, J., Luo, W., Liu, Y., Li, B., Ding, X., Hu, W., ... & Ma, C. (2024). Token Caching for Diffusion Transformer Acceleration. arXiv preprint arXiv:2409.18523.
- [4] Yuan, Z., Zhang, H., Pu, L., Ning, X., Zhang, L., Zhao, T., ... & Wang, Y. (2024). Ditfastattn: Attention compression for diffusion transformer models. Advances in Neural Information Processing Systems, 37, 1196-1219.
- [5] Ma, N., Goldstein, M., Albergo, M. S., Boffi, N. M., Vanden-Eijnden, E., & Xie, S. (2024, September). Sit: Exploring flow and diffusion-based generative models with scalable interpolant transformers. In European Conference on Computer Vision (pp. 23-40). Cham: Springer Nature Switzerland.
- [6] Liu, X., Gong, C., & Liu, Q. (2023, January). Flow Straight and Fast: Learning to Generate and Transfer Data with Rectified Flow. In The Eleventh International Conference on Learning Representations (ICLR).
- [7] Schusterbauer, J., Gui, M., Ma, P., Stracke, N., Baumann, S. A., Hu, V. T., & Ommer, B. (2024, September). FMBoost: Boosting Latent Diffusion with Flow Matching. In European Conference on Computer Vision (pp. 338-355). Cham: Springer Nature Switzerland.
- [8] Lu, C., Zhou, Y., Bao, F., Chen, J., Li, C., & Zhu, J. (2022). Dpm-solver: A fast ode solver for diffusion probabilistic model sampling in around 10 steps. Advances in Neural Information Processing Systems, 35, 5775-5787.
- [9] Lu, C., Zhou, Y., Bao, F., Chen, J., Li, C., & Zhu, J. (2025). Dpm-solver++: Fast solver for guided sampling of diffusion probabilistic models. Machine Intelligence Research, 1-22.

---

> ### Author Rebuttal · Authors · 2025-07-30
>
> # To Reviewer rhnM
> We are pleased that you found our paper well-organized and appreciated the comprehensive experimental evaluation across various generation tasks. Your recognition of the effectiveness of our proposed modules is greatly appreciated. We are committed to addressing any remaining issues and questions you have.
> ## **Weakness 1: Lack of baselines**
> Thank you for emphasizing the importance of comparing our method with strong baseline techniques.
> 1. **Compare with TokenCache [3].**
> | Method | Base Model | FID Score | Speed Up |
> | :---:|:---:| :---:|  :---:|
> |TokenCache|DiT-XL/2 | 2.37 (+0.13) | x 1.32 |
> |Ours| DiT-XL/2 |2.38 **(+0.11)**|**x 1.87**|
>
> We adopt the standard DiT baseline 2.27, while TokenCache applies a baseline 2.24. TokenCache achieves a 32% speed gain with a slight FID increase. Our approach attains an 87% speed gain with a smaller FID increase, demonstrating superior efficiency and performance.
>
> 2. **Compare with Ditfastattn [4].**
> | Method | Base Model | FID Score | IS | Speed Up |
> | :---:| :---: | :---:|:---:| :---:|
> | Ditfastattn | DiT-XL/2 (512x512) | 4.52 (+1.36) |  180.34  | x 1.98|
> |  Ours | DiT-XL/2 (512\x512) | 2.96 **(-0.08)** |**242.4**  | **x 2.45** |
>
> We adopt the standard DIT baseline 3.04, while TokenCache applies a baseline 3.16. Ditfastattn shows a substantial FID increase for speed gain, while our method results in better speed and less FID.
>
> 3. **Compare with DeepCache [2] and BlcokCaching [1]**
>
> DeepCache and BlockCaching share similar features across sampling steps, relying on the redundancy present in multiple steps. In contrast, our approach targets reducing redundancy within the DiT structure at a single sampling step. Our method and their techniques improve diffusion model efficiency along orthogonal dimensions and should be considered complementary rather than directly comparable. For a 1000-step DiT model, DeepCache can achieve substantial reductions in FLOPs—up to 90%—by leveraging massive redundant features across steps. However, obtaining a 90% reduction through model structure compression alone is impractical. Conversely, in a single-step DiT model, our approach can achieve a 50% reduction in FLOPs, while DeepCache and BlockCaching cannot reduce FLOPs, as there are no features to share across steps. Therefore, our method complements DeepCache and BlockCaching by optimizing different aspects of the model's efficiency. Other state-of-the-art methods like DyDiT, DiffCR, and Ditfastattn similarly do not compare against DeepCache and BlockCaching directly.
>
> Instead of comparing solely, we have integrated our approach with DeepCache, demonstrating enhanced performance as follows:
> | Method | interval | FID Score | Speed Up|
> |:---:|:---:| :---:|:---:|
> |DiT-XL | 0 | 2.27 |x 1.0|
> |DiT-XL + DeepCache |2|2.48|x 1.93|
> |DiT-XL + DeepCache + SparseDiT |2|2.78| x 3.52|
>
> These results show the effectiveness of combining SparseDiT with DeepCache.
>
> ## **Weakness 2: Potential mischaracterization of flow-based methods.**
> Thank you for identifying conceptual inaccuracies in our paper. We agree with your points and will revise the flow-based approach descriptions, particularly those related to Rectified Flow, to better reflect their roles and frameworks. We experiment 10-step DPM in Table 5. DiT FID: 12.01; Our FID: 12.30.
>
> ## **Weakness 3: Lack of analysis protocol.**
> (1) SDTM Module Configuration: The number of SDTM modules is empirically determined based on the baseline model's layers. Typically comprising six layers with three sparse transformers, which can achieve a good performance-efficiency trade-off, a DiT-XL houses 28 layers, necessitating four SDTM modules in SparseDiT-XL. Please refer to my response to Reviewer kysk Q2 for more details.
>
> (2) We calculate attention variance to assess the uniformity of attention value distribution. A variance close to zero indicates that attention values are nearly identical, allowing self-attention to function as a global pooling mechanism. Conversely, high variance suggests that each token focuses on a limited set of other tokens. Unlike understanding tasks, such as classification, where global tokens might exist, generation tasks involve attention patterns that tend to capture local information, visible as diagonal lines on attention maps (approximate full-rank matrix). High variance indicates that all tokens are necessary for feature integration. Therefore, a higher variance reflects a greater need for localized information capture. In understanding tasks, high variance can still signify global information capture, but this scenario does not apply to generation tasks.
>
> ## **Weakness 4: Missing isolated analysis of SDTM vs DiT block.**
> It seems there may be a misunderstanding regarding our method. We do not incorporate additional cross-attention layers; rather, the cross-attention layers in SDTM are derived from the existing self-attention layers. The total number of attention layers in SparseDiT, encompassing both self-attention and cross-attention, remains identical to that of DiT. The parameters of the cross-attention layers match those of the initial self-attention layers, hence no additional parameters are introduced via the cross-attention layers. We acknowledge that two additional linear layers, $W_1$ and $W_2$, introduce extra parameters. However, our poolingformers counterbalance this by utilizing parameter-free full-one matrices to replace self-attention layers, thereby reducing overall parameters. Consequently, the parameter count of SparseDiT remains nearly unchanged. Notably, one SDTM comprises six transformer layers. Below, we provide a detailed parameters and FLOPs for these six transformer layers ($r\in$ [0.61, 0.86]):
>
> | Block Name | Parameters | FLOPs|
> | :---:| :---:| :---:|
> | Sparse Token Generation Transformer (Cross-Attention) | 23.905M | 1.539G |
> |  Sparse Transformer (Self-Attention) | 23.905M | 1.023G |
> |  Sparse Transformer (Self-Attention) | 23.905M | 1.023G |
> |  Sparse Transformer (Self-Attention) | 23.905M | 1.023G |
> |  Dense Token Recovery Transformer (Cross-Attention) | 26.562M | 4.258G |
> |  Dense Transformer (Self-Attention) | 23.905M | 4.087G |
>
> Each SDTM consists of six transformers, and should be compared with six DiT transformer blocks in terms of parameters, FLOPs, and throughput:
> | Block Name | Parameters | FLOPs | Throughput|
> |:---:| :---: |:---:|:---:|
> | SDTM | 0.146G | 12.967G | 12.1 |
> | Six DiT Transformers | 0.143G | 24.522G | 7.33 |
>
> Owing to the efficiency of poolingformers, the difference in parameters between SparseDiT and DiT is minimal (680M vs. 680M). The efficiency gains are truly from token sparsity.
>
> ## **Weakness 5: Confound metrics from finetuning.**
> | Model | Continued Iteration | FID |
> | :---:| :---: | :---:|
> | DiT-XL/2 | 400K |  2.15 |
> |  PixArt-$alpha$ | 400K | 4.36 |
>
> We acknowledge that continued training can enhance the performance of base models, as demonstrated in the table above. However, the primary objective of our method is to achieve a superior performance-efficiency trade-off, rather than improving performance. Additionally, the benefits gained from continued training apply equally to all post-training methods, such as DyDiT, DiffCR [10], TokenCache [3], and Ditfastattn [4]. Consequently, the comparative advantages of our method remain consistent.
>
> ## **Q1: High norm tokens**
> We did not specifically observe the norm of tokens in our analysis. Our approach focuses on reducing the number of tokens without directly discarding them. Tokens are merged via pooling layers, which represents a form of soft pruning, allowing both high and low norm tokens to be retained without information loss.
>
> ## **Q2: Upsampling and floating point numbers of $r$**
> Our method utilizes trilinear upsampling, implemented through torch.nn.Upsample. The computation of $r$ is detailed in line 166, where it is adjusted by altering the number of sparse tokens, set in configurations like $4\times 4$ or $8\times 8$. Notably, $r$ remains a discrete value rather than a continuous value.
>
> ## **Q3: l266ff**
> We have observed that our method generates more stable and precise images across different seeds. The seed used in Figure 4 is the default seed 1, chosen to facilitate easy reproduction by readers. We plan to provide additional visualizations in the next version to further support this observation.
>
> ## **Minors**
> We appreciate your detailed suggestions regarding the minors. We will modify these problems in the next version.
>
> ## **Limitation**
> Due to constraints on text space, I have not included a detailed discussion of the solution. Regarding the number of PoolingFormer layers, Figure 1 illustrates that the attention maps from the first two transformer layers exhibit a uniform distribution. Therefore, one or two PoolingFormer layers are adequate, as confirmed by Table 4. In the SDTM architecture, there are three fixed transformers: one for generating sparse tokens, one for recovering dense tokens, and one dense transformer. The number of sparse transformers can be varied, with more sparse transformers enhancing model efficiency. Performance remains consistent as long as the number of sparse tokens does not exceed three. Concerning the pruning rate $r$, our method is not particularly sensitive except for the rate in the final stage. As mentioned in lines 333-336, the effectiveness of the dynamic pruning strategy is closely related to the token count in the later phases of denoising. For practical applications, we suggest using a $10\times 10$ token count in the final stage for optimal performance and a $4\times 4$ token count in the initial stage for efficiency.
>
> ### **Reference**
> [10] You, H., Barnes, C., Zhou, Y., Kang, Y., Du, Z., Zhou, W., ... & Lin, Y. C. (2025). Layer-and Timestep-Adaptive Differentiable Token Compression Ratios for Efficient Diffusion Transformers. CVPR (pp. 18072-18082).

---

> > ### Comment · Reviewer_rhnM · 2025-08-05
> >
> > Thank you for clarifying and addressing some of my concerns, especially weakness 1 (lack of baselines) and weakness 4 (missing isolated analysis of SDTM vs DiT block). I have two more remaining questions:
> >
> > - Q2: I am still confused by the notation. In line 166 you write that $r = 1 - M/N$, with $M \ll N$, which results in a continuous value $r \in [0, 1]$. How can this "remain a discrete value rather than a continuous value"? Especially since with weakness 4 you write $r \in [0.61, 0.86]$ which also seems to be continuous?
> > - Q3: As already mentioned in the review, can you provide an explanation of why your method gives a "more precise structure"? Can you provide at least some support for this claim that goes beyond pure observation?

---

> ### Author Response · Authors · 2025-08-05
> **To Reviewer rhnM**
>
> Thank you very much for your continued engagement and thoughtful feedback！We sincerely appreciate the time and effort you have dedicated to evaluating our rebuttal and the constructive questions you have raised. Your insights are invaluable to us and play a crucial role in enhancing the clarity and rigor of our research. We are committed to addressing your concerns comprehensively and are eager to ensure our paper reaches its highest potential.
>
> Q2: We apologize that our explanation caused any confusion. The variable $M$ represents the number of sparse tokens, which are obtained through pooling operations. Those tokens have discrete values, such as $4\times 4$, $5\times 5$, or $6\times 6$. The $N$ is a fixed, so the equation $r=1-M/N$ results in discrete values. The number of $r$ is limited rather than spanning continuously between 0 and 1. Due to our dynamic pruning rate strategy, the pruning rate $r$ increases as the denoising process progresses. For example, $r\in [0.61, 0.86]$ signifies the the maximum token count is $10\times 10$l yielding $r=1-10\times 10/256=0.61$, while the minimum token count is $6\times 6$, resulting in $r = 1-6\times 6/256=0.86$. During intermediate timesteps, token counts are $7\times 7$ and $8\times 8$, with corresponding $r$ of $0.81$ and $0.75$, respectively. Thus, $r$ belongs to the discrete set {0.61, 0.75, 0.81, 0.86}, which we summarized as $r\in [0.61, 0.86]$ for convince. We apologize again for any misunderstanding and will offer a more detailed explanation in future revisions.
>
> Q3: We think "more precise structure" stems from enhanced global information capture within our model. Sparse tokens encapsulate global information and, due to their limited quantity, integrate this information more effectively than the original DiT. With every token accessing global information, generated structures appear more precise and coherent. Previous research also suggests that global information capture can augment the structural integrity of generated content. For instance, LayoutDiffusion [1] demonstrates the benefits of incorporating global conditions into image features.
> ***
> You said I have addressed some of your questions, especially weakness 1 and weakness 4. How about weakness 2, weakness 3, and weakness 5? Especially weakness 5, We believe it might be a significant concert for you. Please let us know if you have any further questions. We are eager to engage in further discussions and identify any aspects for improvement.
> ***
> ### Reference
> [1] Zheng, G., Zhou, X., Li, X., Qi, Z., Shan, Y., & Li, X. (2023). Layoutdiffusion: Controllable diffusion model for layout-to-image generation. In Proceedings of the IEEE/CVF Conference on Computer Vision and Pattern Recognition (pp. 22490-22499).

---

> > ### Comment · Reviewer_rhnM · 2025-08-05
> >
> > Thank you for clarifying these remaining questions. Clarifying Q2 in the paper might be helpful. Regarding Q3: I still think it's difficult to claim this without further (quantitative) evidence. However, as you addressed my other questions and the mentioned weaknesses I am open for raising my score.

---

> > > ### Author Response · Authors · 2025-08-06
> > > **To Reviewer rhnM**
> > >
> > > Thank you very much for your thoughtful review and for considering raising the score. Thank you once again for your support and encouragement.

---

### Official Review · Reviewer_pfYM · 2025-07-01

**Clarity:** 3
**Significance:** 3
**Originality:** 3
**Rating:** 4
**Confidence:** 4

**Summary:**

The paper introduces SparseDiT, a novel framework designed to tackle the significant computational inefficiency of Diffusion Transformers. The core contribution lies in a unified spatio-temporal token sparsification strategy, motivated by the different attentions shown in different layers and denoising steps. Spatially, it utilizes a tri-segment architecture: a lightweight Poolingformer captures global features in early layers, an innovative Sparse-Dense Token Module (SDTM) alternates between sparse and dense tokens to efficiently process mid-level features, and standard Transformers refine details in final layers. Temporally, it employs a dynamic pruning strategy, progressively increasing the number of active tokens throughout the denoising process to match the evolving demand for detail, thereby optimizing computation across both space and time.

**Questions:**

1.	Temporal sparsification design: Could you provide more justification—either theoretical or empirical—for the choice of T/4 as the divide point of the denoising stage? Why was the specific linear interpolation chosen, and have you considered or tested alternative scheduling strategies?
2.	Limited baseline comparisons: The main paper includes only ablation-type comparisons (with/without SparseDiT), which makes it hard to gauge the method’s performance relative to prior work. Could you include more detailed quantitative and qualitative comparisons with relevant baselines such as ToMeSD, DyDiT, and other token sparsification or merging techniques on more complex  datasets?
3.	Related work inclusion: Several optimization strategies mentioned in the Related Work section (e.g., adaptive token routing, dynamic masking, or hybrid dense-sparse attention methods) are not covered in the experiments. What is the rationale for excluding these, and how do you expect SparseDiT would compare to them?

**Ethical Concerns:**

["NO or VERY MINOR ethics concerns only"]

**Final Justification:**

The rebuttal addresses several of my concerns, particularly by providing more comprehensive baseline comparisons and additional supporting results. However, the core design choices, especially the temporal sparsification schedule, remain primarily empirically driven with limited theoretical grounding. While the paper is technically sound and demonstrates clear practical value, the lack of stronger theoretical support for these mechanisms prevents me from giving a higher score. Given its solid engineering contributions and the potential to inspire follow-up work, I find a borderline accept to be the most appropriate rating. I appreciate the authors’ efforts in the rebuttal and look forward to seeing further refinement and clearer justification of these design choices in future work.

**Limitations:**

A key limitation of this work also stated in limitation part is its heavily manual design choices across both spatial and temporal sparsification modules. Many core architectural decisions—such as the pruning schedule over diffusion steps, the sparse-dense transition structure,—are introduced without theoretical grounding or extensive empirical search, making them appear heuristic rather than principled. This raises concerns about the generality and robustness of the design. Another notable limitation is the insufficient engagement with prior work. While some related methods are briefly mentioned in the Related Work section, the paper does not clearly acknowledge how existing techniques (e.g., token merging from ToMeSD) have inspired or relate to the proposed method. A deeper and more explicit comparison, both conceptual and empirical, would significantly strengthen the work’s positioning and clarify its contribution boundaries.

**Paper Formatting Concerns:**

None formatting issue

**Quality:**

2

**Strengths And Weaknesses:**

The paper derives its core contribution from a careful empirical and conceptual analysis of attention distributions across Transformer layers in DiT, revealing meaningful structural patterns that inform the proposed sparsification mechanism. This insight leads to a well-designed spatial token compression strategy, which is both intuitive and effective, and clearly contributes to improving computational efficiency. The implementation is elegant and shows strong alignment between observation and design. However, the proposed temporal sparsification—adjusting the pruning rate based on diffusion timesteps—feels more arbitrary and lacks sufficient theoretical or empirical grounding. It appears less principled compared to the layer-wise design. A more critical issue lies in the experimental evaluation: the main text only includes with/without SparseDiT comparisons, while the appendix contains limited and coarse comparisons with ToMeSD and DyDiT. Many related methods mentioned in the Related Work section—such as other token pruning, merging, or flow strategies—are not included in the empirical evaluation at all. This makes it difficult to assess the actual advantage and novelty of SparseDiT in the broader context of efficient models.

---

> ### Author Rebuttal · Authors · 2025-07-28
>
> # To Reviewer pfYM
> Thank you for your thoughtful and encouraging feedback. We are delighted that you found value in our empirical and conceptual analysis.
> ## **Q1: Temporal sparsification design.**
> Thank you for this insightful question. We will address it from three perspectives:
> 1. **Theoretical support to timestep-wise pruning rate.** The rationale behind the pruning schedule across different timesteps is detailed in lines 56-58 of our paper. Figure 1b shows that attention variance increases as the denoising process progresses, highlighting a growing emphasis on local information at lower-noise stages. As the denoising proceeds, the model increasingly focuses on local details, dynamically adapting to heightened demand for detail in the later stages. Consequently, we employ larger pruning rates ($r$) in the early stages and smaller rates in the later stages.
>
> 2. **The choice of T/4.** This decision is primarily informed by our experimental findings. We discuss pruning rates in Table 7, and lines 333-334 highlight that the performance of the dynamic pruning strategy is particularly impacted by the token count in the later stages of denoising. Our experiments reveal that performance correlates most strongly with the last T/4 timesteps. Choosing T/3 would entail higher computational costs for tiny gains, and choosing 5/T would result in a noticeable performance decline. T/4 provides the optimal performance-efficiency trade-off.
>
> 3. **Linear interpolation chosen.** Our ablation study, referenced in lines 328-336, provides insights into this decision. Experiments indicate that the performance of the dynamic pruning strategy is particularly sensitive to the token count in the later stages of denoising. Our method is less sensitive to $r$ during early stages. Thus, more complex interpolation and scheduling strategies did not result in significant improvements. We opted for simple linear interpolation to maintain a gradual increase in $r$.
>
> We acknowledge that our paper may not fully convey the rationale behind various choices for our timestep-wise pruning rate strategy, and we will enrich the relevant sections in future revisions.
>
> ## **Q2: Limited baseline comparisons.**
> Thank you for pointing out the need for more comprehensive baseline comparisons. We have focused on comparing our method with ToMeSD and DyDiT because they provide a standardized baseline using DiT. Additionally, DyDiT is recognized as a state-of-the-art technique in this domain. Other methods utilize different base models, and even when employing DiT, they may not adhere to the standard and optimal results from the DiT paper (e.g., FID score 2.27 on $256\times 256$ ImageNet and FID score 3.04 on $512\times 512$ ImageNet), which complicates achieving a perfectly fair comparison. Below, I outline comparisons with nearly all relevant baselines:
>
> 1. **Compare with TokenCache [1].**
> | Method |Base Model|FID Score|Speed Up|
> |:---:|:---:|:---:|:---:|
> | TokenCache | DiT-XL/2 | 2.37 (+0.13) | x 1.32 |
> |  Ours | DiT-XL/2 | 2.38 **(+0.11)** | **x 1.87** |
>
> We adopt the standard DiT baseline 2.27, while TokenCache applies a baseline 2.24. TokenCache achieves a 32% speed gain with a slight FID increase. Our approach attains an 87% speed gain with a smaller FID increase, demonstrating superior efficiency and performance.
>
> 2. **Compare with Ditfastattn [2].**
> | Method | Base Model | FID Score | IS | Speed Up |
> | :---:| :---: | :---:|:---:| :---:|
> | Ditfastattn | DiT-XL/2 (512x512) | 4.52 (+1.36) |  180.34  | x 1.98 |
> |  Ours | DiT-XL/2 (512\x512) | 2.96 **(-0.08)** |   **242.4**  | **x 2.45** |
>
> We adopt the standard DiT baseline 3.04, while TokenCache applies a baseline 3.16. Ditfastattn shows a substantial FID increase for speed gain, while our method results in better speed and less FID.
>
> 3. **Compare with DiffCR [3].**
> | Method | Base Model | FID Score | Speed Up |
> |:---:| :---:|:---:|:---:|
> | DiffCR | PixArt-sigma | 10.68 (**-1.27**) | x 1.26 |
> |  Ours | PixArt-alpha | 4.29 (-0.24) | **x 1.69** |
>
> While DiffCR achieves a greater performance improvement (1.27 vs. 0.24), our results are based on a more challenging baseline of 4.53 compared to their 11.95. The marginal effect implies that enhancing performance from 4.53 is more challenging than from 11.95. Additionally, our method provides a significantly higher speed improvement than theirs.
>
> 4. **Compare with ToMeSD and DyDiT.**
>
> **(1) Quantitative comparisons**
>
> We include quantitative comparisons on class-conditional image generation on ImageNet in Appendix. ToMeSD, being a workshop paper, does not demonstrate high effectiveness, whereas DyDiT delivers substantial results. Therefore, I mainly compare our results with DyDiT. For the text-to-image task, we both employ PixArt-$alpha$ as our base model. DyDiT uses a smaller dataset, COCO, while we utilize a more complex dataset, SAM. The results are:
>
> | Method | Base Model | FID Score ($\Delta$) | Speed Up |
> |:---:|:---:|:---:|:---:|
> | DyDiT | PixArt-alpha | 19.75 (-0.13) | x 1.32 |
> |  Ours | PixArt-alpha | 4.29 (**-0.24**) | **x 1.69** |
>
> Our SparseDiT demonstrates better performance improvement and speed up in the text-to-image task.
>
> **(1) Quantitative comparisons**
>
> Due to OpenReview's limitations on image uploads, we assess visual quality using ChatGPT's evaluation comparing our SparseDiT with ToMeSD and DyDiT:
>
> **ToMeSD (Model A) v.s.  SparseDiT (Model B)**
>
> ChatGPT assessment:
>
> Model B is a far more powerful, capable, and useful generative model than Model A.
>
> Profile of Model B (Bottom Set):
>
> Technically Superior: It consistently produces high-resolution, sharp, and richly detailed images.
>
> Aesthetically Driven: It demonstrates a sophisticated understanding of composition, lighting, and color theory to create beautiful, engaging results.
>
> Idealized Generation: It excels at capturing the core essence of a concept (e.g., the eruption of a geyser, the majesty of a mountain) and presenting it in its most ideal and visually striking form.
>
> Profile of Model A (Top Set):
>
> Resembles a "Raw Database": Its output mirrors the varied quality and style of a large, unfiltered image dataset scraped from the web, containing both good and bad examples.
>
> Occasional Serendipity: Due to its less guided nature, it can sometimes produce unique and narratively rich images (like the recumbent parrot) that a more polished, aesthetically-driven model might not.
> For any practical application requiring high-quality, visually appealing, and reliable image generation, Model B is the clear and definitive winner.
>
> **DyDiT (Model A) v.s.  SparseDiT (Model B)**
>
> Both our Appendix and DyDiT's provide numerous visualizations, with our images presented at a resolution of
> $512\times 512$ in Figures 4-15 and DyDiT's at $256\times 256$ in Figures 9-22. Despite the difference in resolution, the comparison remains valid as both utilize the same training dataset. While both sets of results exhibit high generation quality, SparseDiT images display a more precise structure in certain cases compared to DyDiT.
>
> ChatGPT assessment:
>
> Model A (left) is a model that strives for ultimate aesthetics. It will create the most stunning, idealized, and dramatic images, sometimes at the cost of coherence. Its "ceiling" is very high, capable of producing breathtaking results. However, its "floor" is low; the artifact in the parrot image proves that it can fail in certain cases and cannot be fully trusted.
>
> Model B (right) is a model that prioritizes realism and robustness. Its output is more like a faithful reproduction of high-quality photographs from the real world. Its aesthetic "ceiling" might not be as stunning as Model A's best shots, but its "floor" is very high—it rarely, if ever, produces obvious errors, making its output highly reliable and consistent.
>
> ## **Q3 Related work inclusion**
> Several optimization strategies mentioned in our related work. We have listed some comparisons in Q2. Some methods, like flow-based methods and timestep distillation methods focus on reducing or optimizing timesteps to enhance efficiency, whereas our method centers on pruning model structure and reducing FLOPs within a single timestep. Our method and their techniques improve diffusion model efficiency along orthogonal dimensions and should be considered complementary rather than directly comparable. In Table 5, we have shown that our method can be integrated with advanced sampling methods utilizing fewer timesteps. Additionally, we present another experiment to showcase its combination with DeepCache:
>
> | Method | interval | FID Score | Speed Up|
> |:---:|:---:| :---:|:---:|
> |DiT-XL | 0 | 2.27 |x 1.0|
> |DiT-XL + DeepCache |2|2.48|x 1.93|
> |DiT-XL + DeepCache + SparseDiT |2|2.78| x 3.52|
>
> If there are specific papers or methods you believe warrant further comparison, I would appreciate you point out them and we will do further comparisons.
>
> ## **Limitations**
> (1) For hyperparameter selection details, please refer to my response to Reviewer kysk Q2.
>
> (2) I provide an analysis of ToMeSD in our Appendix. Unlike ToMeSD, which directly adapts motivation and techniques from vision transformers, our approach involves a detailed observation of DiT's behavior in generation tasks and a comprehensive redesign of all modules accordingly.
> ### **Reference**
> [1] Lou, J., Luo, W., Liu, Y., Li, B., Ding, X., Hu, W., ... & Ma, C. (2024). Token Caching for Diffusion Transformer Acceleration. arXiv preprint arXiv:2409.18523.
>
> [2] Yuan, Z., Zhang, H., Pu, L., Ning, X., Zhang, L., Zhao, T., ... & Wang, Y. (2024). Ditfastattn: Attention compression for diffusion transformer models. NeurIPS, 37, 1196-1219.
>
> [3] You, H., Barnes, C., Zhou, Y., Kang, Y., Du, Z., Zhou, W., ... & Lin, Y. C. (2025). Layer-and Timestep-Adaptive Differentiable Token Compression Ratios for Efficient Diffusion Transformers. CVPR (pp. 18072-18082).
>
> [4] G. Fang, X. Ma, and X. Wang. Structural pruning for diffusion models. NeurIPS, 2023.

---

> > ### Comment · Reviewer_pfYM · 2025-08-05
> >
> > Thank you for the authors’ response. As a reviewer, I carefully examined the rebuttal and believe the following points remain worth discussing:
> >
> > For Question 1, the authors’ explanation is primarily based on empirical results and ablation studies. While this partially addresses my concerns and improves the interpretability of the design, the theoretical justification could be further strengthened.
> >
> > For Question 2, the response is generally sufficient and substantially enhances the baseline comparisons. Given the strict constraints on image presentation in NeurIPS rebuttal, the use of AI-assisted qualitative image descriptions is an interesting attempt. However, the results produced by ChatGPT remain subjective and lack reproducibility. I would recommend that the authors provide more comprehensive visualizations in the camera-ready version to improve the persuasiveness of their results.
> >
> > Regarding Question 3, the authors’ explanation of complementarity and the additional experiments are reasonable. Nevertheless, I suggest that the paper could further elaborate on the potential integration of the proposed method with other approaches, to better clarify its contribution and applicability.
> >
> > Overall, I believe my initial score remains appropriate. I appreciate the authors’ efforts during the rebuttal stage, which have resolved part of my concerns. I look forward to future work in this direction, which may inspire further research and development in the field.

---

> > > ### Author Response · Authors · 2025-08-05
> > > **To Reviewer pfYM**
> > >
> > > Thank you very much for your thoughtful feedback and for carefully reviewing our rebuttal. Your insights are invaluable and greatly appreciated as we work to improve our paper.
> > >
> > > ### **For Q1:**
> > > We concur with your observation that the theoretical justification could be further strengthened. Recent studies [1, 2, 3, 4, 5] indicate that diffusion models display varying denoising behavior over sampling steps. They generate the low-frequency global structure information in the early denoising stage and the high frequency details in the late denoising stage. Consequently, the token count requirements increase progressively along the sampling steps. Moreover, the cascaded diffusion model [6] employs a cascading pipeline wherein a base model first generates $32\times 32$ images, followed by two super-resolution models producing $64\times 64$ and $128\times 128$ images, respectively. This method can be considered another way to reduce tokens. Our dynamic pruning rate strategy also can be seen as a enhanced cascaded diffusion model. Each phase provides well-initialized low-resolution image for subsequent phase, thus, requiring fewer sampling steps to complete the denoising processing. Our experiment show that each phase only need 1/4 steps for the task.
> > >
> > > ### **For Q2:**
> > > We agree with the view of reviewer that the results produced by ChatGPT remain subjective and lack reproducibility. We have to apply this method due to the regular of NeurIPS. We will provide more visualizations in the camera-ready version to improve the persuasiveness of their results.
> > >
> > > ### **For Q3:**
> > > We apologize for not fully understanding your question. I mistakenly believed that our focus should solely be on comparing our method with others in our experiments. In fact, you are suggesting that our method achieve potential integration of the proposed method with other approaches.
> > >
> > >
> > > Our method is indeed capable of being combined with other techniques. In Table 5, we illustrate the effective combination of our method with efficient samplers, demonstrating synergy in scenarios with a few sample steps. Additionally, during the rebuttal, we have integrated our method with the training-free approach, DeepCache, which shares features across adjacent sampling steps. This result is presented in response to Q3. We are committed to exploring further integrations with approaches like adaptive token routing, dynamic masking, or hybrid dense-sparse attention methods, and hope to present more findings in our next version.
> > > ***
> > > We will advance our research in this area and hope our future work will contribute to further developments and innovations in the field. Your insights will play an essential role in guiding us.
> > >
> > > ### **Reference**
> > > [1] Jooyoung Choi, Jungbeom Lee, Chaehun Shin, Sungwon Kim, Hyunwoo Kim, and Sungroh Yoon. Perception prioritized training of diffusion models. In Proceedings of the IEEE/CVF Conference on Computer Vision and Pattern Recognition, pages 11472–11481, 2022.
> > >
> > > [2] Jonathan Ho, Ajay Jain, and Pieter Abbeel. Denoising dif- fusion probabilistic models. Advances in neural information processing systems, 33:6840–6851, 2020.
> > >
> > > [3] Hengyuan Ma, Li Zhang, Xiatian Zhu, and Jianfeng Feng. Accelerating score-based generative models with preconditioned diffusion sampling. In European Conference on Computer Vision, pages 1–16. Springer, 2022.
> > >
> > > [4] Robin Rombach, Andreas Blattmann, Dominik Lorenz, Patrick Esser, and Bjo ̈rn Ommer. High-resolution image synthesis with latent diffusion models. In Proceedings of the IEEE/CVF conference on computer vision and pattern recognition, pages 10684–10695, 2022.
> > >
> > > [5] Xingyi Yang, Daquan Zhou, Jiashi Feng, and Xinchao Wang. Diffusion probabilistic model made slim. In Proceedings of the IEEE/CVF Conference on computer vision and pattern recognition, pages 22552–22562, 2023.
> > >
> > > [6] Ho, J., Saharia, C., Chan, W., Fleet, D. J., Norouzi, M., & Salimans, T. (2022). Cascaded diffusion models for high fidelity image generation. Journal of Machine Learning Research, 23(47), 1-33.

---

> > > > ### Comment · Reviewer_pfYM · 2025-08-06
> > > >
> > > > Thank you for your response. Good luck!

---

### Official Review · Reviewer_kysk · 2025-07-01

**Clarity:** 3
**Significance:** 3
**Originality:** 3
**Rating:** 5
**Confidence:** 5

**Summary:**

In this paper, the authors aim to improve the DiT method by pruning its architecture. They propose a new architecture called SparseDiT, which achieves performance comparable to the original DiT while using significantly fewer FLOPs after fine-tuning pretrained DiT models. The main approach is to: (1) apply average pooling to extract sparse tokens, (2) alternate self-attention on the sparse tokens with cross-attention between the sparse and dense tokens, and (3) apply a few dense attention layers at the end. In a fine-tuning setting, the authors show that their proposed method can achieve performance comparable to DiT while significantly reducing FLOPs.

**Questions:**

1. As raised in the weaknesses section, how does the model perform when trained from scratch? Is its effectiveness limited to a fine-tuning context, or can it achieve strong performance without relying on weights from a pre-trained dense model?

2. Could the authors provide more insight into the selection of the model's hyperparameters?

(a) Is there a principled method for determining the number of layers for each module (e.g., PoolingFormer, SDTM)? How sensitive is the model's performance to this specific configuration?

(b) Similarly, what is the rationale behind the design of the pruning schedule across different timesteps (t)? How sensitive is the final performance to changes in this schedule?

3. The model's non-uniform architecture raises questions about potential optimization challenges. Conventional transformers benefit from well-studied optimization strategies due to their uniform layer structure. Does the heterogeneity of SparseDiT introduce any new difficulties for training stability? For instance, when training from scratch, do the different modules require distinct learning rates or optimizers to converge effectively?

**Ethical Concerns:**

["NO or VERY MINOR ethics concerns only"]

**Final Justification:**

I thank the authors for their rebuttal. The newly presented results on pretraining the proposed architecture are impressive and compelling. This evidence significantly strengthens the paper's contribution, demonstrating that the proposed architecture has the potential to be highly impactful for the design of future diffusion-based models.

Given these strong results, which solidify the paper's contribution to the field, I now recommend acceptance.

**Limitations:**

Yes

**Quality:**

3

**Strengths And Weaknesses:**

Strengths

1. In Figures 1 and 3, the authors show that the early-layer attention maps in DiT are highly uniform. This is an insightful observation that provides a clear motivation for using average pooling.

2. The authors validate the proposed SparseDiT across various benchmarks, including image and video generation, demonstrating improved efficiency in all cases. The experiments are solid.

3. The performance gains appear consistent across different model sizes, showing strong potential for scalability.

Weaknesses

1. My major concern is with the experimental setup. The authors only report performance after fine-tuning a pre-trained model and do not include results from training SparseDiT from scratch. The claims would be much stronger if the proposed architecture also performs comparably to the original DiT when pre-trained. Otherwise, the method should be compared to knowledge distillation techniques, which also create smaller, more efficient models from a pre-trained teacher.

2. The architectural design is non-uniform and involves many hyperparameters, including:

(1) The number of layers for the PoolingFormer, the sparse-dense modules, and the final dense block.

(2) The specific pruning ratio schedule used across different timesteps (t).

I am concerned that without a principled way to determine these hyperparameters, the method's broader applicability may be limited.

---

> ### Author Rebuttal · Authors · 2025-07-27
>
> # To Reviewer kysk
> Thank you for your valuable feedback and for highlighting the strengths of our work. We are delighted that you found our observations regarding early-layer attention maps in DiT insightful and supportive of the use of average pooling. We appreciate your recognition of the comprehensive validation of SparseDiT across various benchmarks and your acknowledgment of the experiments as solid. Furthermore, your observation of consistent performance gains across different model sizes indicates the scalability potential of our approach.
>
> We are committed to addressing each of your inquiries and suggestions thoroughly in the upcoming revisions. Your feedback is instrumental in enhancing the quality of our research.
> ## **Q1: Training from scratch.**
> Thank you for your insightful observation. Our method indeed can be trained from scratch; however, leveraging a well-pretrained model offers significant advantages, aligning with our research goal of enhancing the efficiency of well-trained diffusion models by pruning tokens. Unlike specialized architectural design methods such as Pixart-$\alpha$ [1], our aim is not to develop a new model architecture. We just argue that our findings can inspire future designs. We opted for fine-tuning rather than training from scratch for two primary reasons:
>
> 1. Computational Cost: Training diffusion models from scratch, such as our baseline model, DiT-XL/2, requires 7,000K iterations, approximately two months on A100 GPUs. While our training speed surpasses the original DiT, it still demands about a month of processing, which is not feasible for us. Especially considering more complex T2I models, their computational cost will be more tremendous.
>
> 2. Practical Application: From a practical standpoint, an algorithm that requires training from scratch holds less value compared to one that can be fine-tuned. Fine-tuning techniques are increasingly common and effective. Leading open-source foundation generation models such as WanX2.1, HunyuanVideo, and Flux are built on traditional DiT architectures and derive their efficient variants through fine-tuning. Comparable works [2, 3, 4] also employ fine-tuning or training-free strategies without training from scratch.
>
> I understand the importance of assessing our model's capability and the dependency of pre-trained weights. We have trained our model from scratch, although due to time constraints, only reaching 400K training iterations during the rebuttal period. A complete training cycle like DiT requires 7,000K steps, roughly equivalent to one month of computing. We compare our result with original DiT paper in Table 4. The results (cfg=1.0) are as follows:
> | Model | FID | FLOPs | Iterations |
> | :---: | :---: | :---: |  :---: |
> | DiT | 19.47 | 118.64 | 400K |
> | Ours (from scratch) | 15.11 | 68.05 | 400K |
>
> These results indicate our method can be trained from scratch, achieving strong performance without dependence on pre-trained dense model weights. Nevertheless, we believe our method’s fine-tuning capability possesses more practical significance.
>
> Regarding comparisons with distillation techniques, diffusion models commonly use timestep distillation, with some methods claiming single-step inference [5, 6, 7]. Unlike these, our approach focuses on model pruning to reduce computational costs in one step. Our method and distillation techniques improve diffusion model efficiency along orthogonal dimensions and should be considered complementary rather than directly comparable. For a one-step distillation method, they can reduce the timestep from 100 to 1, which saves 99\% FLOPs, whereas achieving a 99\% reduction in FLOPs through model structure compression alone is nearly impossible. To our knowledge, related works also do not compare their results with distillation methods.
>
> ## **Q2: The selection of the model's hyperparameters.**
> 1. We have conducted ablation studies, as shown in Tables 4 and 6, to demonstrate the selection of several key hyperparameters. Regarding the number of PoolingFormer layers, Figure 1 indicates that the attention maps from the first two transformer layers display a uniform distribution. Consequently, one or two PoolingFormer layers suffice, as corroborated by Table 4. In the SDTM architecture, one transformer is designated for generating sparse tokens, another for recovering dense tokens, and one is dense transformer. These three transformers remain fixed. We provide an additional ablation study in the below table to verify it.
> The number of sparse transformers can be adjusted, with more sparse transformers yielding more efficient models. Performance remains stable as long as the number of sparse tokens does not exceed three. The final dense block should contain no more than three transformers to ensure efficiency, with additional transformers having no significant impact on performance.
> | No. of sparse transformer | FID | FLOPs |
> | :---: | :---: | :---: |
> | 3 | 2.38 | 68.05G |
> | 4 | 2.76 | 69.41G |
> | 7 | 3.89 | 65.67G |
> 2. The rationale for the pruning schedule across different timesteps is detailed in lines 56-58. In Figure 1 b, we find that attention variance increases as the denoising process advances, indicating a growing emphasis on local information at lower-noise stages. As denoising progresses, the model increasingly prioritizes local details, dynamically adapting to heightened demands for detail at later stages. Therefore, we adopt larger pruning rate at early stages while smaller pruning rate at later stages. We discuss how to set the pruning rates in Table 7. Our method is not sensitive for the pruning rates beside the pruning rate of the last stage. As said in line 333-336, the performance of the dynamic pruning strategy is particularly relative with the token count of the later stages of denoising. For instance, the configuration with token counts ranging from $4\times 4$ to $8\times 8$ only shows a 0.02 FID score increase compared to the configuration with a constant $8\times 8$ token count. In the practical application, we recommend a $10\times 10$ token count in the final stage for optimal performance, and a $4\times 4$ token count in the first stage for efficiency. The intermediate stages can be set to $6\times 6$ and $8\times 8$, with the method displaying robustness to these settings.
>
> I will reorganize the content above and include it in our supplementary material to clearly assist readers in selecting the hyperparameters.
>
> ## **Q3: Training stability.**
> Despite our model's non-uniform architecture, it demonstrates remarkable training stability. PoolingFormer structures are inherently stable, as evidenced by [8]. Our SDTM architecture alternates between sparse and dense structures, with dense transformers ensuring stability. The final top block comprises standard transformers, which naturally exhibit stability. Throughout extensive experiments, we have encountered minimal issues concerning training stability. The sole instance of instability arises when no PoolingFormer layers are used, causing a sudden shift to sparse tokens in the initial layer. In training DiT-XL/2 from scratch, we employed a singular global learning rate, achieving rapid convergence. The training loss decreased from 0.0308 at 100 iterations to 0.0120 at 10,000 iterations, affirming the model's stability.
> ### **Reference**
> [1] Chen, J., Yu, J., Ge, C., Yao, L., Xie, E., Wu, Y., ... & Li, Z. (2023). Pixart-$\alpha $: Fast training of diffusion transformer for photorealistic text-to-image synthesis.
>
> [2] You, H., Barnes, C., Zhou, Y., Kang, Y., Du, Z., Zhou, W., ... & Lin, Y. C. (2025). Layer-and Timestep-Adaptive Differentiable Token Compression Ratios for Efficient Diffusion Transformers.
>
> [3] Zhao, W., Han, Y., Tang, J., Wang, K., Song, Y., Huang, G., ... & You, Y. (2024). Dynamic diffusion transformer.
>
> [4] Bolya, D., & Hoffman, J. (2023). Token merging for fast stable diffusion.
>
> [5] Yin, T., Gharbi, M., Zhang, R., Shechtman, E., Durand, F., Freeman, W. T., & Park, T. (2024). One-step diffusion with distribution matching distillation.
>
> [6] Song, Y., Dhariwal, P., Chen, M., & Sutskever, I. (2023). Consistency models.
>
> [7] Xie, S., Xiao, Z., Kingma, D., Hou, T., Wu, Y. N., Murphy, K. P., ... & Gao, R. (2024). Em distillation for one-step diffusion models.
>
> [8] W. Yu, M. Luo, P. Zhou, C. Si, Y. Zhou, X. Wang, J. Feng, and S. Yan. Metaformer is actually what you need for vision.

---

> > ### Comment · Reviewer_kysk · 2025-08-05
> >
> > Thanks the authors to the clarifications. The authors fully address my concerns on the stability of training from scatch, and the sensitivity to the hyperparameters. I decided to raise my score to Accept.

---

> > > ### Author Response · Authors · 2025-08-05
> > > **To Reviewer kysk**
> > >
> > > Thank you very much for your invaluable insights. We sincerely appreciate the dedication and effort you have invested in reviewing our work. Your questions has been instrumental in guiding us towards enhancing the quality and impact of our paper.

---

### Official Review · Reviewer_gHMa · 2025-07-01

**Clarity:** 3
**Significance:** 3
**Originality:** 3
**Rating:** 4
**Confidence:** 4

**Summary:**

The paper introduces SparseDiT, a novel framework for enhancing the computational efficiency of Diffusion Transformers (DiT) through token sparsification across spatial and temporal dimensions. The authors employ a tri-segment architecture (Poolingformer, Sparse-Dense Token Modules, and dense transformers) to dynamically adjust token density, achieving significant reductions in FLOPs (up to 55%) and improvements in inference speed (up to 175%) while maintaining generative quality, as demonstrated on tasks like image and video generation.

**Questions:**

Please see the weakness.

**Ethical Concerns:**

["NO or VERY MINOR ethics concerns only"]

**Final Justification:**

This work proposed to speedup DiT with further finetuning. The results are quite positive. However, there are still concerns on hurting initial pretraining distribution, which could hinder the performance of the pretrained model.

**Limitations:**

yes

**Quality:**

3

**Strengths And Weaknesses:**

## Strengths

1. Clear motivation. The token sparsification strategy is well-motivated by empirical observations of attention patterns in DiT layers.

2. The paper demonstrates impressive performance gains across multiple benchmarks (e.g., ImageNet, video datasets, and text-to-image generation) with minimal degradation in output quality (e.g., FID).

3. Overall, this paper is easy to follow and is also written well.


## Weaknesses

1. My main concern is the practical value of this technique. Specifically, today's text to image/video generation models are pretrained on large-scale datasets with many parameters. Thus, finetuning makes it rather challenging as it may change the underlying distribution of their initial generation space. For this reason, training-free methods seem to be more likely preferred by the community.

2. Lack deeper analysis of sparse/dense tokens. For example, in L301-302, the paper claims that "Sparse tokens capture global structures, while dense tokens capture detailed information.", however, it would be more convincing if the authors can provide some visualizations or comparisons, e.g. more sparse tokens help to capture global or more dense tokens help to generate detailed information.

---

> ### Author Rebuttal · Authors · 2025-07-26
>
> # To Reviewer gHMa
> Thank you for recognizing the strengths in our paper, including the well-motivated token sparsification strategy and our impressive performance across multiple benchmarks. We're glad the paper was clear and well-written. We are eager to address your queries and improve our work based on your valuable feedback.
> ## **Q1: The practical value of this technique.**
> Training-free methods indeed garner significant interest in the community. These methods often target redundant sampling steps in diffusion models. For instance, advanced sampling strategies like DDIM and DPM can achieve similar results in fewer steps compared to DDPM, while DeepCache shares features between adjacent steps. Our approach, however, emphasizes a different dimension—simplifying the model structure, which typically necessitates additional fine-tuning. Specifically, we prune the number of tokens in a single sampling step, and thereby cannot circumvent finetuning. As diffusion models evolve, the number of sampling steps decreases, reducing the scope for optimization within sampling steps. We think that the significance of efficient model structures will increase.
>
> Fine-tuning is widely recognized as an essential method for achieving good performance-efficiency diffusion models. Many common methods often require additional fine-tuning. For example, standard RF [1], consistency models [2], and other sampling step distillation [3] techniques all necessitate this process. Leading open-source T2I model Flux [4], and T2V model HuanyuanVideo [5] both employ additional distillation for efficiency gains. Consequently, we argue that minimal fine-tuning for enhancing efficiency is justifiable in practical applications.
>
> Moreover, the overhead associated with fine-tuning in our approach is reasonable. We require only 6% of the training iterations. For instance, DiT-XL/2 necessitates just 400K iterations for fine-tuning (about 1.5 dayes on 8 A100 GPUs), compared to 7,000K iterations for full training, achieving comparable efficacy to recent works [6, 7, 8].
>
>
> ## **Q2: Lack deeper analysis of sparse/dense tokens.**
> Thank you for pointing out this crucial aspect, which warrants further exploration.
>
> The differentiation between global and local tokens arises from observations documented in Figure 1, where some tokens correspond to larger areas of images, while others focus on smaller regions. Our method harnesses and amplifies this phenomenon by directing certain layers to generate global tokens, while others preserve their original representation. To substantiate the effectiveness of our approach, we extract the attention map with dimensions N$\times$ N, where N is the number of tokens of an H$\times$W image. By reshaping the attention map to $N\times H \times W$, each $H\times W$ heatmap is the receptive field of the corresponding token. It is evident that global token heatmaps emphasize larger regions, presenting more precise and meaningful semantic information, whereas local token heatmaps concentrate on smaller areas. I apologize for OpenReview's current constraints with image uploads, preventing us from displaying these visualizations. Instead, I provide quantitative results, calculating normalized variances of heatmaps using the method from Figure 1b, detailed in the table below:
>
> | TimeStep | Global Token | Local Token |
> | :---: | :---: | :---: |
> | 10 | 0.29 | 1.00 |
> | 500 | 0.18 | 0.82 |
> | 900 | 0.00 | 0.68 |
>
> Note: Values are normalized to a 0-1 scale, aligned with Figure 1b. Higher variance indicates a local property, while a lower variance suggests a global property. This table demonstrates that global and local tokens effectively capture global and local information, respectively. We intend to include visualization results in the next version of our paper.
>
>
> ### **Reference**
> [1] Liu, X., Gong, C., & Liu, Q. (2022). Flow straight and fast: Learning to generate and transfer data with rectified flow.
>
> [2] Song, Y., Dhariwal, P., Chen, M., & Sutskever, I. (2023). Consistency models.
>
> [3] Yin, T., Gharbi, M., Zhang, R., Shechtman, E., Durand, F., Freeman, W. T., & Park, T. (2024). One-step diffusion with distribution matching distillation. computer vision and pattern recognition (pp. 6613-6623).
>
> [4] FLUX.1: Redefining Text-to-Image AI with Superior Visual Fidelity.
>
> [5] Kong, W., Tian, Q., Zhang, Z., Min, R., Dai, Z., Zhou, J., ... & Zhong, C. (2024). Hunyuanvideo: A systematic framework for large video generative models.
>
> [6] You, H., Barnes, C., Zhou, Y., Kang, Y., Du, Z., Zhou, W., ... & Lin, Y. C. (2025). Layer-and Timestep-Adaptive Differentiable Token Compression Ratios for Efficient Diffusion Transformers. Computer Vision and Pattern Recognition Conference (pp. 18072-18082).
>
> [7] Zhao, W., Han, Y., Tang, J., Wang, K., Song, Y., Huang, G., ... & You, Y. (2024). Dynamic diffusion transformer. ICLR.
>
> [8] Yuan, Z., Zhang, H., Pu, L., Ning, X., Zhang, L., Zhao, T., ... & Wang, Y. (2024). Ditfastattn: Attention compression for diffusion transformer models. Advances in Neural Information Processing Systems, 37, 1196-1219.

---

### Comment · Area_Chair_xrJm · 2025-08-03

Dear Reviewers,

Thanks for your hard work during the review process. We are now in the author-reviewer discussion period.

Please (1) carefully read all other reviews and the author responses; (2) start discussion with authors if you still have concerns as early as possible so that authors could have enough time to response; (3) acknowledge and update your final rating. Your engagement in the period is crucial for ACs to make the final recommendation.

Thanks,

AC

---

> ### Comment · Area_Chair_xrJm · 2025-08-05
>
> Dear Reviewers gHMa, pfYM, and rhnM,
>
> As we're approaching the end of author-reviewer discussion period, please read the rebuttal and start discussion with the authors as soon as possible. If all your concerns have been addressed, please do tell them so. Please note that submitting mandatory acknowledgement without posting a single sentence to authors in discussions is not permitted. Please also note that __non-participating reviewers will receive possible penalties of this year's responsible reviewing initiative and future reviewing invitations.__
>
> Thanks,
>
> AC

---

### Note · Authors · 2025-08-14

Dear Chairs and Reviewers,

Thank you to all the reviewers for their insightful and encouraging feedback on our paper. We are grateful for the recognition of the clear motivation behind our token sparsification strategy, especially the observations of attention patterns in DiT (Reviewer gHMa and Reviewer kysk). Our impressive performance gains across benchmarks, with minimal degradation in output quality, is deeply appreciated and noted by Reviewer gHMa and Reviewer kysk. We also value Reviewer rhnM's recognition of our comprehensive experimental evaluation and the simplicity yet effectiveness of our proposed modules.

The discussions during the rebuttal phase with reviewers were immensely valuable, and we are pleased to have addressed most of their questions. We express our gratitude for the positive scores given by all reviewers, especially to Reviewer kysk for raising the score to Accept. Contributions of our work include: (1) pioneering a design specifically aimed at accelerating Diffusion Transformer models through token sparsification; (2) implementing token sparsification across both spatial and temporal dimensions for enhanced computational efficiency; (3) conducting extensive experiments to demonstrate the effectiveness of our methods in image generation, video generation, and T2I generation. We believe this work can significantly develop scalable and efficient diffusion models for high-resolution content generation, expanding the practical applications of DiT models.

Through this rebuttal process, our work has become more complete and solid. We presented more comprehensive baseline comparisons and verified our state-of-the-art performance in this field, acknowledged by Reviewer pfYM and Reviewer rhnM. Additionally, we demonstrated our method's compatibility with other efficient approaches, noted by Reviewer pfYM and Reviewer rhnM. We highlighted the ability to train our model from scratch, in response to Reviewer kysk. Furthermore, we refined the strategy for selecting hyperparameters, acknowledged by Reviewer kysk, Reviewer pfYM, and Reviewer rhnM. We also provided deeper analysis of sparse and dense tokens, as suggested by Reviewer gHMa.

Looking forward, we plan to include more baseline comparisons, diverse visualizations, and deeper analyses of our proposed methods, along with conclusions, such as "more precise structure". **Moreover, we will release our code in the future to further facilitate research and collaboration.**

Best,

Authors

---

### Decision · Program_Chairs · 2025-09-17

**Decision:**

Accept (poster)

**Comment:**

This paper presents a novel framework for improving the computational efficiency of diffusion Transformers through token sparsification. Reviewers acknowledged the contribution and strong performance of the proposed method, while initially raising some concerns, such as its unclear experiment setups and incomplete baseline comparisons.

After the rebuttal, the authors addressed most of the concerns, and all reviewers agreed to accept this paper. AC read all the reviews, author rebuttals, and the paper, and believes this is a strong paper and recommends acceptance.